# Orchestrated Sparse Consortium of Small Experts Beats Monolithic LLMs

## Abstract

Large Language Models (LLMs) attain impressive capabilities but demand heavy computation and offer limited transparency. Naively shrinking a model reduces computational overhead yet typically sacrifices breadth and performance; we, therefore, pursue a different axis: keep models modular and *scale up* by coordinating multiple experts such that a small, task-adaptive subset collaborates per input and could achieve better performance. In this paper, we introduce FOCUS (**F**lexible **O**rchestration and **C**ollaboration **U**sing **S**pecialists) – a *generic* multi-expert collaboration framework that trains a lightweight *orchestrator* under *oracle* supervision to *select, order,* and *coordinate* a consortium of *experts* (homogeneous/heterogeneous language models of any size). A learnable sparse, near-symmetric *collaboration matrix* governs information flow among experts, and a *multi-round refinement* process aggregates intermediate outputs into a single answer; the oracle is only used during training, not at test time. During test time, the orchestrator adaptively routes the experts with early stopping, achieving only *sublinear cost growth* in terms of the consortium size. FOCUS achieves striking results: on MMLU, GSM8K, and HumanEval, a consortium of 5-7 Qwen experts (combined ∼9B parameters) reach 94.1%, 94.1%, and 87.8% accuracy, respectively, matching or surpassing a Qwen3-14B model by an average margin of 7.6%. On reasoning benchmarks, a consortium of 5 Phi-4-Mini improves AIME-2024 from 26% to 40% and GPQA-DIAMOND from 19% to 31%, and attains 92% on MATH-500, exceeding a single Phi-4-14B reasoning model. These results establish *collaboration* as a distinct axis of scaling: carefully orchestrated experts can outperform comparable-size monolithic models while remaining modular and cost-effective for deployment.

## 1 Introduction

Large, monolithic language models achieve state-of-the-art results across a wide range of tasks, but their sheer scale makes them costly to train and deploy, latency-prone at inference, and difficult to interpret (Kaplan et al., 2020; Hoffmann et al., 2022; Zhao et al., 2024). A complementary direction is to *coordinate a consortium of models* (*aka* **consortium** of **experts**), potentially from different families and sizes, so that a small, carefully chosen subset of experts collaborate per input. Prior lines of work point to the promise (and limits) of such coordination: sparsely-activated mixture-of-experts expands capacity at near-constant per-token compute (Lepikhin et al., 2020; Fedus et al., 2022), while ensemble and multi-agent schemes (e.g., debate, rank-and-fuse, and agent frameworks) can improve reliability but often keep a large mediator in the loop or rely on hand-crafted interaction rules (Du et al., 2023; Jiang et al., 2023; Wu et al., 2024; Shinn et al., 2023; Wang et al., 2022). Motivated by these observations and by the empirical gap between single compact models and much larger ones, we adopt a *generic* view: collaboration should not be restricted by scale. The core question we study is: **Can a small consortium of experts, orchestrated effectively, match or outperform a larger monolithic model at lower cost and higher efficiency?**

Several lines of prior work explore multi-expert or multi-agent frameworks, but each has limitations. An emerging paradigm is to use a powerful Large Language Model (LLM) to *coordinate* other models. For example, an orchestration model might decide which expert to consult for each question or might prompt multiple LLMs to debate or discuss. These multi-expert prompting frameworks have demonstrated improved accuracy and truthfulness through techniques like debate (Du et al., 2023) or via fusing answers post-hoc (Jiang et al., 2023). Liu et al. (2024) proposed a dynamic

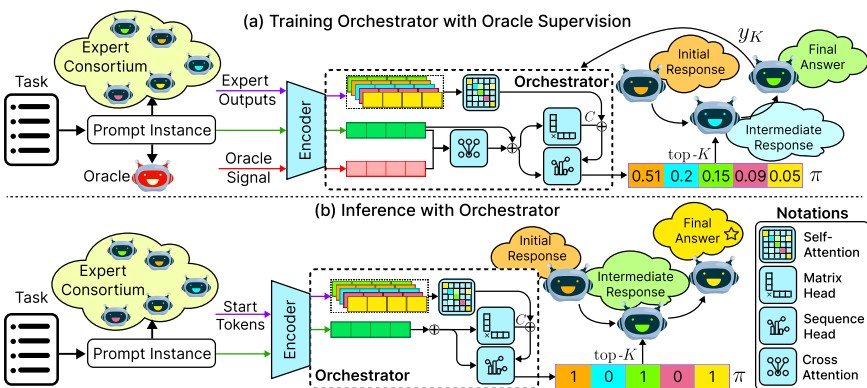

Figure 1: **An overview of FOCUS: (a) Training.** For an input prompt $x$ and available *consortium* of *experts* $\mathcal{E}_i$s, the *orchestrator* encodes expert embeddings along with *oracle* signal $o$, and outputs a *sequence distribution* $\pi$ to select/order experts. Top-$K$ experts refine in sequence $y_0 \to y_1 \to \cdots \to y_K$ under losses that enforce the structure of *collaboration*. **(b) Inference.** The oracle is absent; the orchestrator deterministically invokes $\text{Top}-K(\pi)$ with *early stopping* when edits are redundant, yielding sublinear inference-time cost. Experts are frozen; only the orchestrator is trained.

LLM-powered expert network for task-oriented collaboration in which a large model adaptively builds and updates a network of specialized experts, deciding how they should interact for a given task. However, they often require a large model "in the loop" to mediate at inference time (for instance, acting as a judge or as a router to construct and manage the network), which reintroduces the deployment cost of an LLM. They also tend to rely on hand-crafted interaction rules (e.g., turn-taking in a debate or static voting), rather than a learned protocol, and thus may not fully exploit the potential of collaboration. Yang et al. (2025b) introduced a decentralized, evolutionary mechanism where experts iteratively adapt their communication strategies without a central orchestrator. This reduces inference-time dependence on a large orchestrator but requires costly evolutionary search.

Here, we introduce FOCUS (**F**lexible **O**rchestration and **C**ollaboration **U**sing **S**pecialists) – a novel, generic and modular collaboration framework among a consortium of experts (see Figure 1). In the framework, a small (typically sized $\approx$ 1M) **orchestrator**, trained under **oracle** supervision, produces (i) a *collaboration matrix* that governs directed information flow among experts and (ii) a *sequence distribution* that selects and orders a sparse subset of experts for each input. Inference proceeds via *multi-round refinement*: the first selected expert proposes a solution; subsequent experts condition on prior outputs and refine them; an *early-stopping* rule halts when further edits are redundant. This design applies uniformly whether experts are small, medium, or large, and whether the consortium is homogeneous (same model family) or heterogeneous (different model families). Our extensive experiments across complex reasoning benchmarks like MMLU, GSM8K, HumanEval, AIME and GPQA reveal striking results and highlight a prominent generic scaling relation. • **A consortium of small experts beats monoliths:** For a $K$-expert system with combined size of $B$, we observe that the performance of the multi-expert system $\mathcal{A}_{\text{FOCUS}}(K, B) \gtrsim \mathcal{A}_{\text{mono}}(\rho(K)\,B)$, the performance of a monolithic model of size $\rho(K)\,B$, with typical $\rho(K) \in [1.4, 1.6]$. For instance, on GSM8K, a consortium of 8-12 Qwen3-1.7B or 4B experts achieves nearly 20% better accuracy than the base models, closing most of the gap to Qwen3-32B. On MATH-500, a consortium of three reasoning-tuned Phi-4-Mini-3.8B models even *surpasses* a single Phi-4-14B reasoning model. • **Smaller experts reap better benefits:** We empirically observe a scaling behavior $\mathcal{A}_{\text{FOCUS}}(K, B) \approx 100 - \gamma(\frac{B}{K})K^{-\alpha(\frac{B}{K})}$, with both $\gamma(\frac{B}{K})$ and $\alpha(\frac{B}{K})$ being monotonically decreasing functions of $\frac{B}{K}$, the average expert size. This empirical inverse scaling behavior hints at the higher effectiveness of FOCUS on smaller language models (see Figure 2); for instance, a Qwen3-1.7B model boosts its single-model accuracy by over 10% with just 3 experts, whereas a multi-expert system built from Qwen3-8B models yields only marginal improvements with a similar number of experts. • **FOCUS favors sparser consortium:** We further observe that number of activated experts (*aka* cost) in the consortium follows a power-law trend: $\mathcal{C}_{\text{FOCUS}}(K, B) \approx 1 + \delta(\frac{B}{K})K^{-\beta}$, with $\beta \approx 0.5$ and $\delta(\frac{B}{K}) \in (0, 0.5)$ an increasing function of average expert size. For instance, in a consortium of 14 Qwen3-1.7B experts, only 2.1 experts are activated for a given instance, whereas, for a consortium of 5 Qwen3-8B experts, 2.3 experts are activated on average.

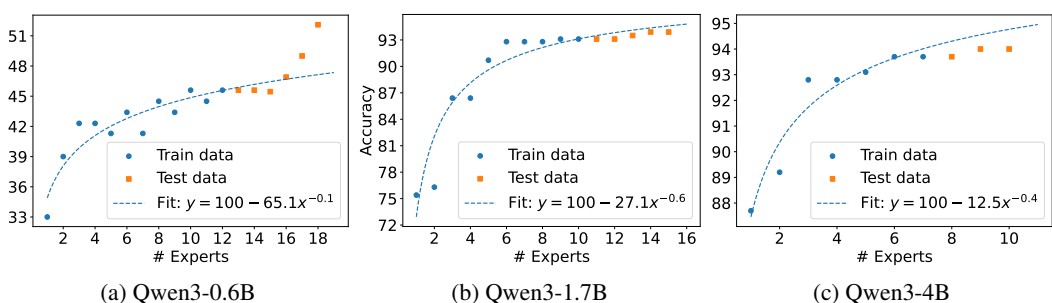

(a) Qwen3-0.6B       (b) Qwen3-1.7B       (c) Qwen3-4B

Figure 2: Scaling of FOCUS on GSM8K: for models $\leq$ 4B, using multiple experts boosts performance by up to 20% over the single-expert baseline.

In summary, our contributions are as follows[1]: (1) We present a generic multi-expert framework that learns a *differentiable collaboration protocol* – a collaboration matrix for directed refinement and a selection policy for sparse activation, guided by an oracle only during training. (2) We introduce structural objectives (sparsity and near-symmetry) and an oracle-free inference procedure with early stopping that together provide efficiency at test time. (3) We highlight empirical scaling behaviors for coordinated experts and demonstrate that a carefully orchestrated consortium can match or surpass larger monolithic models across knowledge, math, and code-generation tasks, defining a favorable accuracy-cost Pareto frontier. (4) We introduce a robust, scalable, cross-family capable method to realize meaningful learned orchestration where existing multi-expert approaches struggle. A detailed list of **FAQs** related to FOCUS and their responses are provided in Appendix A.

## 2    RELATED WORK

**Multi-expert frameworks driven by collective reasoning.** Several frameworks have been proposed to make LLMs cooperate on tasks. Debate frameworks let multiple language models (LMs) argue or critique each other's answers to improve accuracy (Irving et al., 2018; Du et al., 2023). For instance, LLM-Debate uses multiple models to solve math problems, with a judge (human or model) picking the final output (Du et al., 2023). While effective for truthfulness, such methods need many expensive queries and a judging step, making them computationally costly. Other approaches (Estornell & Liu, 2024; Khan et al., 2024) show the promise of collective reasoning but depend on rigid protocols and all-expert participation, limiting scalability.

**Orchestrator-driven multi-expert systems.** Another family of methods uses a mediator or planner LLM to coordinate others. The "LLM-as-manager" approach (Shinn et al., 2023) decomposes tasks for specialists, while systems like Autogen (Wu et al., 2024) enable expert interaction under a central orchestrator. These improve reasoning and coding but require a strong LLM in the loop. LLM-Blender (Jiang et al., 2023) instead uses a ranker to pick candidate answers before fusing them, but such methods treat models as black boxes, combining outputs rather than enabling deeper collaboration. Ensemble baselines (majority or confidence-weighted voting, self-consistency (Wang et al., 2022)) boost robustness but cannot create new reasoning paths. Input-based selectors like DyLAN (Liu et al., 2024) reduce cost by activating subsets of experts, conditioned only on input embeddings, in isolation. These yield limited gains because the routing decisions are limited to input-level features and ignore the complexity of the task and possible inter-expert interactions.

**Network driven multi-expert systems.** Recent work formalizes multi-LLM systems as networks of interacting agents. GPTSwarm (Zhuge et al., 2024) models LMs as nodes in a communication graph optimized via evolutionary algorithms, while others explore reinforcement learning (RL) or game-theoretic coordination (Park et al., 2023). These approaches aim to structure multi-model interactions but often optimize topologies or prompts in ad hoc ways rather than learning principled protocols. AgentNet (Yang et al., 2025b) proposes a self-organizing graph where experts delegate tasks without a central orchestrator, though it struggles in heterogeneous settings. In a similar manner, Liu et al. (2025) cast collaboration as a multi-expert RL problem, introducing Multi-Agent GRPO to optimize expert cooperation.

---

[1]We include FOCUS source code in the supplementary material and will open-source it upon acceptance.

**Why is FOCUS unique?** In contrast, our FOCUS framework learns an explicit, differentiable collaboration protocol under oracle supervision without needing to train or fine-tune the underlying experts for the collaboration. The orchestrator produces a sparse and symmetric collaboration matrix governing information flow and a sequence distribution that activates only a small subset of experts, enabling multi-round refinement where experts improve upon each other's outputs. Unlike debate, voting, or fusion-based baselines, we achieve higher accuracy with fewer activated experts, offering a more efficient and principled route to multi-expert collaboration.

## 3 METHODOLOGY

**Overview.** FOCUS trains a lightweight *orchestrator* under *oracle* guidance *during training*. Given a consortium of $M$ experts $\{\mathcal{E}_i\}_{i=1}^M$, the orchestrator produces two objects per input: (i) a *collaboration matrix* $C \in \mathbb{R}^{M \times M}$ encoding directed handoff probabilities among experts, and (ii) a *sequence distribution* $\pi \in \Delta^M$ scoring which experts to activate. At inference, a small subset (top-$K$ by $\pi$) is executed as a multi-round refinement chain guided by $C$ with early stopping. Experts are always frozen; only the orchestrator is trained. The design choice of FOCUS ensures several key properties: (i) **near-symmetric collaboration:** ($C \approx C^\top$), promoting reciprocal and stable handoffs between experts; (ii) **sparse consortium:** via row-entropy, yielding few strong edges and interpretable routing; (iii) **cost-aware:** combining a length penalty with early stopping to keep realized chains short (typically 2-3 experts) and compute sublinear. Figure 1 summarizes training and inference. All notations used in the methodology are elaborated in Table 5 of Appendix B.

**Input encoding and contextualization.** For each input, expert $\mathcal{E}_e$ produces an embedding $o_e \in \mathbb{R}^d$ (via a shared encoder like BERT-base (Devlin et al., 2019)). These are projected to orchestrator space as $h_e = W_{\text{in}} o_e + b$, with $W_{\text{in}} \in \mathbb{R}^{d_h \times d}$ and $b \in \mathbb{R}^{d_h}$. During training we *softly* inject oracle information $o \in \mathbb{R}^d$ by attention $\tilde{h}_e \leftarrow h_e + \text{Attn}(h_e, W_o o, W_o o)$, where $W_o \in \mathbb{R}^{d_h \times d}$. Stacking expert states $R = [\tilde{h}_1; \ldots; \tilde{h}_M] \in \mathbb{R}^{M \times d_h}$, we let experts exchange context via self-attention (Vaswani et al., 2017), $\tilde{R} \leftarrow R + W_{\text{res}} \text{Attn}(W_{\text{in}} R, W_{\text{in}} R, W_{\text{in}} R)$, with $W_{\text{res}} \in \mathbb{R}^{d \times d_h}$. During test time, we only use the non-contextualized representations $h_e$ for deriving the expert state $R$, due to the absence of the oracle, while the self-attention over experts preserves the collaboration inductive bias.

**Learning the collaboration matrix.** The orchestrator turns contextualized expert representations $\{r_i\}$ (rows of $\tilde{R}$) into $C$ by first computing logits $C_{\text{logits}} \in \mathbb{R}^{M \times M}$, then adding a semantic prior that encourages routing among representationally compatible experts:

$$(C_{\text{logits}})_{ij} \leftarrow (C_{\text{logits}})_{ij} + \cos(r_i, r_j), \tag{1}$$

followed by masking self-edges $(C_{\text{logits}})_{ii} = -\infty$ and applying a row-wise softmax, $C_{i \to j} = \frac{\exp\{(C_{\text{logits}})_{ij}\}}{\sum_{j'} \exp\{(C_{\text{logits}})_{ij'}\}}$. Interpreting $C$ as a soft adjacency matrix over experts yields a stable routing graph: rows encode where expert $i$ delegates next; columns reflect how often an expert is selected as a refiner.

**Identifying expert collaboration sequence.** Next, the orchestrator computes a *sequence distribution* $\pi \in \Delta^M$ that assigns each expert a probability of being included in the final response chain. This distribution is derived by combining multiple signals: (i) a base **sequence score** from the input encoding, (ii) an expert **competence score**, (iii) the **collaboration structure**, and (iv) a **length penalty**. For each expert $e$, we calculate a unnormalized selection logit $z_e \in \mathbb{R}^M$ as:

$$z_e = f_{\text{seq}}(h_e) + f_{\text{perf},e}(\tilde{R}) + f_{\text{collab, e}}(C) - f_{\text{len, e}} \tag{2}$$

A two-layer MLP $f_{\text{seq}}(h_e) = W_{\text{seq}} h_e + b_{\text{base}}$ (with $W_{\text{seq}} \in \mathbb{R}^{M \times d_h}$) produces a raw score for each expert based on the encoded prompt. A small positive bias $b_{\text{base}}$ is added to each logit to ensure no expert has zero probability. We then estimate each expert's expected performance via a linear predictor: $f_{\text{perf},e}(\tilde{R}) = w_p^\top r_e$, where $w_p \in \mathbb{R}^d$ is a learned weight vector and $r_e \in \mathbb{R}^d$ is expert $e$'s representation in the shared contextualized representation $\tilde{R}$. This term biases $z$ toward experts that appear more *effective* for the given input. We also incorporate the collaboration structure by defining $f_{\text{collab, e}}(C) \in \mathbb{R}^M$ as the average of the expert's outgoing and incoming edge strengths:

$$f_{\text{collab},e}(C) = \frac{1}{2}\left(\frac{1}{M-1}\sum_{j' \neq e} C_{e \to j'} + \frac{1}{M-1}\sum_{i \neq e} C_{i \to e}\right), \tag{3}$$

which reflects how centrally expert $e$ is connected in the collaboration graph. Finally, to discourage excessive reliance on later-indexed experts (which could increase latency), we impose a length-based penalty: $f_{\text{len},e} = \alpha\, e + \beta$ for the $j$-th expert (assuming experts are indexed in a fixed order $1, \ldots, M$ sorted by increasing expected inference speed). Here, $\alpha$ and $\beta$ are small positive constants. A minimal penalty $\beta$ is ensured, and each subsequent expert has a slightly higher penalty. This pushes the selection towards using fewer experts and preferring earlier (faster) experts when possible. We obtain $\pi = \text{GumbelSoftmax}(z; \tau)$ during training to enable differentiable sampling; at test time, we select the top-$K$ experts by $z$ (deterministic). Crucially, $\pi$ decides *who* participates, while $C$ specifies *in which order* they should refine one another by following high-probability transitions.

**Multi-round expert interaction.** Once the orchestrator selects a sequence $S = (j_1, \ldots, j_K)$, the input is processed in multiple rounds. The first expert $\mathcal{E}_{j_1}$ generates an initial output from the prompt. Each subsequent expert $\mathcal{E}_{j_t}$ receives both the original input and the previous output $y_{t-1}$, while collaboration weights $C_{j_{t-1} \to j_t}$ modulate how strongly it attends to earlier experts. This produces a chain of refinements, culminating in the final prediction $y_K$. Thus, $C$ not only determines which experts are active but also governs the flow of information across them, turning independent models into a coordinated reasoning pipeline.

**Training objective and optimization.** Only the orchestrator is trainable; experts remain frozen. For each example with a hard label $y^*$, we minimize a weighted sum of components:

- **Utility** $\mathcal{L}_{\text{utility}}$ (task loss on $y_K$ vs. $y^*$): anchors training to the end-task so that all collaboration ultimately improves the *final* prediction, not merely intermediate consistency.

- **Distillation** $\mathcal{L}_{\text{distill}}$ $\|o - o_{\mathcal{E}_K}\|_2^2$: encourages the final expert to mimic the oracle's reasoning by minimizing the mean-squared error between their respective hidden representations. This component acts as a stabilizer and helps accelerate convergence, especially for small consortia.

- **Symmetry** $\mathcal{L}_{\text{symm}} = \|C - C^\top\|_F^2$: discourages brittle, one-way topologies in the collaboration graph. Near-symmetric $C$ promotes *reciprocal* information flow, which empirically yields shorter, more stable refinement chains and better robustness to population changes (e.g., when an expert is removed or replaced). Ablations confirm this is a critical structural prior, preventing performance degradation in larger consortia.

- **Sparsity** ($\mathcal{L}_{\text{spar}}$): encourages a sparse collaboration matrix by penalizing the average entropy of its rows. For each row $C_i$ (distribution of expert $i$'s outgoing weights), we compute the entropy $H(C_i) = -\sum_j C_{i,j} \log C_{i,j}$, and penalize its average across all experts. This encourages each expert to rely on a small, selective set of other experts rather than forming diffuse connections.

- **Oracle alignment** $\mathcal{L}_{\text{oracle}} = \frac{1}{M^2}\sum_{i,j}(C_{ij} - C_{ij}^{\text{oracle}})^2 + \frac{1}{M}\sum_i(\pi_i - \pi_i^{\text{oracle}})^2$ with $C_{ij}^{\text{oracle}} = \cos\left(\frac{o_i + o_j}{2}, o\right)$ ($i \neq j$) and $\pi_i^{\text{oracle}} = \cos(o_i, o)$: bootstraps a sensible *initial protocol*. It steers both *who* to select and *how* to connect toward oracle-indicated preferences, then cedes to the utility loss as training progresses. This prevents long cold-start phases where the orchestrator explores unproductive chains. Ablations confirm the oracle acts as an accelerator and stabilizer for early training, with test-time performance driven by the learned protocol.

- **Diversity** $\mathcal{L}_{\text{diver}} = -\frac{1}{M}\text{var}(s_1, \ldots, s_M)$ with $s_i = \sum_{k \in \text{TopK}(\pi)} \pi_k \mathbf{1}[k = i]$: counters *mode collapse* onto a single persistent expert by encouraging balanced utilization across the consortium. This broadens exploration, improves coverage on heterogeneous workloads, and works in tandem with sparsity to allocate distinct niches.

- **Selection entropy** $\mathcal{L}_{\text{sel}} = -\frac{1}{M}\sum_i \pi_i$: encourages *decisive* selections (low entropy), which reduces dithering across many near-tied experts, stabilizes the Gumbel-Softmax path, and shortens realized chains by clarifying the top-$K$.

- **Length penalty** $\mathcal{L}_{\text{len}} = K \cdot \alpha$: encodes the compute budget directly into the objective. It regularizes toward *short* chains, synergizing with early stopping and the length-aware score $f_{\text{len},e}$ to yield favorable accuracy-latency trade-offs.

Therefore, the total loss is

$$
\begin{aligned}
\mathcal{L}_{\text{total}} &= \lambda_{\text{utility}}\mathcal{L}_{\text{utility}} + \lambda_{\text{distill}}\mathcal{L}_{\text{distill}} + \lambda_{\text{symm}}\mathcal{L}_{\text{symm}} + \lambda_{\text{spar}}\mathcal{L}_{\text{spar}} \\
&\quad + \lambda_{\text{oracle}}\mathcal{L}_{\text{oracle}} + \lambda_{\text{diver}}\mathcal{L}_{\text{diver}} + \lambda_{\text{sel}}\mathcal{L}_{\text{sel}} + \lambda_{\text{len}}\mathcal{L}_{\text{len}} ,
\end{aligned}
\tag{4}
$$

optimized with backpropagation. We use the Gumbel-Softmax reparameterization to pass gradients through discrete selections during training.

**Adaptive mechanisms and inference.** Two schedules stabilize training and improve efficiency. First, *temperature annealing* sharpens selections: $\tau \leftarrow \max(\tau_{\min}, \gamma\tau)$ with $\gamma \in (0, 1)$. Second, *adaptive chain length* reduces the allowed $K$ once a validation metric $q(t)$ exceeds a threshold $\theta$: $K \leftarrow \max(K_{\min}, \lfloor \delta K \rfloor)$ with $\delta \in (0, 1)$. At test time, the oracle is removed, $(C, \pi)$ are computed once using minimal start tokens from all experts, then top-$K$ experts by $z$ are executed in the order prescribed by $C$, and early stopping returns the final output when further refinement is redundant. The result is a compact collaboration policy that activates only a few high-value experts per input while preserving the performance benefits of collaboration.

## 4 EXPERIMENTAL SETUP AND BASELINES

We extensively test FOCUS with different model families, including – Qwen-3 (Yang et al., 2025a), Phi-4 (Abdin et al., 2024) and LLaMA-3 (Dubey et al., 2024) with sizes ranging from 0.6B to 32B. The performing tasks include – MMLU (Hendrycks et al., 2020), spanning 57 tasks across diverse domains, high-school mathematical reasoning benchmark GSM8K (Cobbe et al., 2021) and code generation benchmark HumanEval (Chen et al., 2021). Additionally, we experiment with Phi-4 reasoning model on complex reasoning tasks – AIME 2024 (MAA Committee, 2025), AIME 2025, GPQA Diamond (Rein et al., 2024) and MATH-500 (Lightman et al., 2023). For each of these tasks, we train the orchestrator model on the validation split and report the performance on the test split. To construct the expert consortium, we use both homogeneous (replica of same model) and heterogeneous (different pre-trained models from same or different model families), with varying consortium size $M \in [1, 15]$. For all experiments, we use GPT OSS 20B (Agarwal et al., 2025) as the oracle model during training of the framework. We employ BERT-base-uncased (Devlin et al., 2019) as our shared embedding model, for encoding the expert and oracle outputs. The trainable component of the orchestrator is ∼1M parameters in size, with the BERT-base encoder remaining frozen. We highlight the system prompt used with experts, along with illustrations of expert collaboration in Appendix C.

We use Adam optimizer ($\eta = 1 \times 10^{-3}$) with a cosine learning rate scheduler with warmup, and a batch size of 2 for training the orchestrator model for each task. We train the orchestrator for 5 epochs and combine multiple loss terms with the following weights: $\lambda_{\text{utility}} = 0.5$, $\lambda_{\text{distill}} = 0.5$, $\lambda_{\text{symm}} = 0.1$, $\lambda_{\text{sparse}} = 0.01$, $\lambda_{\text{oracle}} = 0.3$, $\lambda_{\text{diver}} = 0.1$, $\lambda_{\text{sel}} = 1.0$, and $\lambda_{\text{len}} = 0.5$. This selection was made through a thorough hyperparameter sweep, elaborated in Appendix D. We set the length cost $\alpha$ and minimum penalty $\beta$ as 0.2 and 0.1, respectively. The hidden dimension in orchestrator, $d_h$ is set to 256. The Gumbel-Softmax temperature $\tau$ is initialized at 1.0 and decayed by 0.999 with $\tau_{\min} = 0.5$. . During inference, we set the early-stopping string similarity threshold to 0.8. All experiments are executed on a single NVIDIA A100 80GB GPU. Detailed decoding parameters for the LLM experts are provided in Appendix C.

We compare FOCUS with three multi-expert baselines – (i) **LLM-Blender** (Jiang et al., 2023) is a static ensemble method that averages predictions across different models, without adaptive routing or dynamic selection. (ii) **LLM-Debate** (Du et al., 2023) organizes multiple experts in an iterative debate protocol, where each expert sequentially refines or critiques prior outputs, but at the cost of invoking all experts for every query. (iii) **DyLAN** (Liu et al., 2024) adopts a more efficient strategy by selecting a subset of experts based on input features to balance accuracy and efficiency.

## 5 RESULTS

**Performance improvement over single-experts.** Tables 1a and 1b compare single-expert baselines with FOCUS. We find that expert collaboration consistently boosts accuracy, often surpassing much larger models. For example, Qwen3-8B with FOCUS achieves an average of **91.2%**, outperforming Qwen3-32B (83.2%) and Qwen3-14B (84.2%), while using only 2–3 experts per query. Mid-scale Qwen3-4B reaches 84.2%, matching Qwen3-14B despite being 4× smaller, and Qwen3-1.7B improves from 64.3% to **83.1%**, rivaling the single-expert Qwen3-8B baseline (78.0%). On GSM8K, gains are particularly large (up to +14.0% for Qwen3-1.7B), while HumanEval also shows strong improvements (e.g., +21.4% for Qwen3-8B). At the smallest scale, Qwen3-0.6B rises from 34.3% to 50.0%, and LLaMA3-8B improves modestly to 63.3%, indicating that very weak experts benefit

less. Overall, FOCUS enables small and mid-sized experts to match or exceed much larger LMs, delivering both accuracy and efficiency gains.

Table 2 reports results when FOCUS is applied to consortia of heterogeneous experts, combining models of different sizes or families. We find that a carefully composed consortium often achieves performance competitive with or exceeding homogeneous ensembles. For example, pairing Qwen3-8B with multiple Qwen3-4B experts yields strong improvements: Qwen3-8B + 3×Qwen3-1.7B attains an average of **89.0%**, surpassing most single model baselines and approaching the best homogeneous Qwen3-8B setup (91.2%). Notably, Qwen3-8B + 2×Qwen3-4B achieves 88.8% average, indicating that augmenting a moderately large expert with smaller ones provides complementary gains, especially on GSM8K where accuracy exceeds 94%. Similarly, Qwen3-4B + 4×Qwen3-1.7B reaches 88.2%, showing that aggregating mid-scale and small experts can rival larger models. Consortia with weaker or less

| Model | MMLU | GSM8K | HumanEval | Average |
|---|---|---|---|---|
| Qwen3-32B | 84.0 | 93.7 | 72.0 | 83.2 |
| Qwen3-14B | 92.9 | 92.6 | 67.0 | 84.2 |
| Qwen3-8B | 77.4 | 89.6 | 67.0 | 78.0 |
| Qwen3-4B | 72.3 | 87.7 | 63.5 | 74.5 |
| Qwen3-1.7B | 55.0 | 75.0 | 62.8 | 64.3 |
| Qwen3-0.6B | 38.7 | 33.0 | 31.3 | 34.3 |
| LLaMA3-8B | 34.2 | 66.7 | 60.0 | 53.6 |
| Phi-4-Mini | 66.3 | 88.6 | 65.8 | 73.6 |

(a) Single-expert results

| Model | MMLU | GSM8K | HumanEval | Average | Δ |
|---|---|---|---|---|---|
| Qwen3-8B | 94.1 (2.3) | 91.0 (1.1) | 88.4 (2.2) | 91.2 | **+13.2** |
| Qwen3-4B | 85.6 (2.1) | 94.0 (3.6) | 73.0 (1.9) | 84.2 | **+9.7** |
| Qwen3-1.7B | 83.5 (2.1) | 93.9 (14.0) | 72.0 (1.8) | 83.1 | **+18.8** |
| Qwen3-0.6B | 56.2 (3.2) | 52.1 (1.7) | 35.6 (1.7) | 50.0 | **+15.7** |
| LLaMA3-8B | 48.5 (3.6) | 77.3 (3.8) | 64.0 (1.0) | 63.3 | **+9.7** |

(b) Multi-expert results with FOCUS

Table 1: Results with different LMs. (a) single-expert baselines. (b) Performance of FOCUS with consortium of **homogeneous** experts (average number of experts activated is shown in parentheses). The Δ column highlights absolute improvements in average score, showing that FOCUS enables small and mid-sized models (e.g., 1.7B, 4B) to rival or surpass much larger baselines (≥14B).

compatible models are less effective: combining Phi-4-Mini and LLaMA3-8B with Qwen3-8B achieves only 78.0% average, significantly below Qwen3-only consortium, highlighting that cross-family collaboration may suffer from representational misalignment. Nevertheless, certain hybrid setups remain competitive, e.g., Qwen3-8B + LLaMA3-8B + Qwen3-4B delivers 81.8%, outperforming its individual constituents. Overall, these results demonstrate that FOCUS effectively leverages heterogeneous consortium, with the best gains arising when experts share some architectural affinity (e.g., within Qwen models), while heterogeneous consortium across families yield more modest benefits due to alignment challenges.

| Model Combination | Total Size (B) | MMLU | GSM8K | HumanEval | Average |
|---|---|---|---|---|---|
| *Consortium with Qwen3-8B* | | | | | |
| Qwen3-8B + Qwen3-4B | 12.0 | 88.2 | 88.6 | 88.4 | 88.4 |
| Qwen3-8B + 2× Qwen3-4B | 16.0 | 88.9 | 88.6 | 89.6 | 88.8 |
| Qwen3-8B + Qwen3-1.7B | 9.7 | 88.2 | 92.8 | 83.5 | 88.2 |
| Qwen3-8B + 2× Qwen3-1.7B | 11.4 | 88.2 | 93.2 | 84.1 | 88.2 |
| Qwen3-8B + 3× Qwen3-1.7B | 13.1 | **89.0** | **94.1** | **87.8** | **89.0** |
| *Consortium with Qwen3-4B* | | | | | |
| Qwen3-4B + Qwen3-1.7B | 5.7 | 78.4 | 86.1 | 76.8 | 82.2 |
| 2× Qwen3-4B + Qwen3-1.7B | 9.7 | 78.4 | 86.1 | 81.7 | 82.2 |
| 2× Qwen3-4B + 2× Qwen3-1.7B | 11.4 | 81.7 | 92.8 | 82.9 | 87.2 |
| 2× Qwen3-4B + 3× Qwen3-1.7B | 13.1 | 81.7 | 93.1 | 82.9 | 87.4 |
| 2× Qwen3-4B + 4× Qwen3-1.7B | 14.8 | 83.4 | 93.1 | 83.5 | 88.2 |
| *Cross-family Consortium* | | | | | |
| Phi-4-Mini + LLaMA3-8B + Qwen3-8B | 20.0 | 80.4 | 74.5 | 79.2 | 78.0 |
| Qwen3-8B + LLaMA3-8B + Qwen3-4B | 20.0 | 88.2 | 78.7 | 78.6 | 81.8 |
| 2× Phi-4-Mini + 2× LLaMA3-8B + 2× Qwen3-8B | 40.0 | 86.1 | 79.6 | 85.6 | 83.8 |

Table 2: Performance of FOCUS with a consortium of **heterogeneous** experts. The *Total Size* column reports the aggregate parameter count (in billions). Qwen-only consortium consistently deliver the strongest accuracy with moderate total size, while cross-family consortium require larger capacity but underperform due to representational misalignment.

**Comparison with other multi-expert baselines.** Table 3 compares FOCUS against existing multi-expert frameworks, including static blends (LLM-Blender), iterative debate methods (LLM-Debate), and adaptive selectors (DyLAN). The results show that while these baselines can provide moderate

| Category | Best Baseline Avg. | FOCUS Improvement | Cost (Eff. Params, B) |
|---|---|---|---|
| Single models | 73.6 | **+17.6/+8.6** | 4.0 |
| LLM-Blender (heterogeneous) | 42.0 | **+49.2/+40.0** | 12.5 |
| LLM-Debate (heterogeneous) | 47.3 | **+43.9/+34.9** | 12.5 |
| LLM-Debate (homogeneous; Phi×7) | 79.5 | **+11.7/+2.7** | 28.0 |
| DyLAN (homogeneous; 6.3/9 Phi models) | 75.6 | **+15.6/+6.6** | 25.2 |
| FOCUS (best accuracy; 1.88/5 active, 5×8B models) | **91.2** | — | 15.4 |
| FOCUS (least cost; 4B+1.7B models) | **82.2** | — | **5.7** |

Table 3: Summary comparison of FOCUS (best) against multi-expert baselines with effective parameter cost. Effective parameter sizes are computed from Table 6 in Appendix D as: Blender/Debate (hetero) query all three models ($8B+4B+0.5B \approx 12.5B$); Debate (homo, Phi) uses $7 \times 4B \approx 28B$; DyLAN(Phi×3) activates $6.4/9$ Phi experts $\approx 25.2B$. For FOCUS, we use consortium of 5 Qwen3-8B models (8.19B each) with 1.88 models active on average. The smallest configuration with FOCUS contains a Qwen-4B and a Qwen-1.7B, totaling 5.7B.

gains over single models, they either plateau in performance or incur substantial computational overhead. LLM-Blender yields only marginal improvements, averaging $42.0\%$, and debate-style ensembles remain inefficient: for example, LLM-Debate with 7 Phi experts achieves $79.5\%$ average accuracy but requires querying all experts at every step ($100\%$ cost). DyLAN improves efficiency by activating a subset of experts, yet its performance lags behind; DyLAN (Phi ×3) reaches $75.6\%$ at $69.7\%$ cost, and DyLAN (LLaMA3-1B × 5) achieves $54.6\%$ at $66.7\%$ cost. In contrast, FOCUS consistently delivers both higher accuracy and lower cost. With just 3 experts, it achieves $83.1\%$ average accuracy, surpassing all DyLAN and debate baselines, while activating only 2.0 experts on average ($68\%$ cost). Increasing to 5 experts raises accuracy to $85.7\%$ with cost dropping further to $46\%$, and at 7 experts FOCUS achieves a state-of-the-art **87.3%** average while using only $34.2\%$ of the available experts. This contrasts sharply with LLM-Debate, which scales poorly, and DyLAN, which cannot match the accuracy despite similar cost levels. Naive multi-expert aggregation methods are highly sensitive to the constitution of the consortium, whereas FOCUS is able to perform consistently in highly imbalanced, as well as, balanced consortia (see Tables 6 and 7) by being able to suppress unhelpful experts and amplify constructive refinements. Overall, FOCUS establishes a new Pareto frontier in multi-expert collaboration: it consistently outperforms baselines in accuracy while simultaneously reducing the fraction of experts activated, demonstrating that intelligent orchestration is superior to both static ensembling and debate-driven redundancy.

**Performance scaling with multiple experts.** Figure 3 shows a consistent pattern across model sizes: accuracy rises sharply with the first few activated experts and then saturates. For Qwen3-1.7B, GSM8K climbs from $75.0\%$ (one expert) to $> 93\%$ with 8-10 experts (exceeding Qwen3-14B); on HumanEval, it moves from $62.8\%$ to $\sim 72\%$ with only 2-3 experts. Qwen3-4B exhibits a similar trend: MMLU grows $72.3\% \to 84.0\%$ by 4-5 experts and GSM8K passes $92\%$ with $\sim 3$ experts, while Qwen3-8B saturates early (MMLU $> 94\%$ at $K=3$; HumanEval $\sim 88\%$ at $K=2$-3). This behavior is well captured by the parametric law $y(x) = 100 - \alpha\, x^{-\beta}$, which models diminishing returns with the number of experts $x$. Fits on GSM8K (Figure 2) yield low extrapolation errors for 0.6B/1.7B/4B/8B experts ($2/0.5/0.7/0.6\%$), indicating robust predictability. In short: collaboration scales accuracy quickly, saturates predictably, and due to early stopping, keeps realized cost modest.

We measure *cost* as the *average number of activated experts* in a consortium. With early stopping (*c.f.* Figure 7 of Appendix D) and a length penalty, realized cost grows *sublinearly*: most gains arrive by $K \in [2, 4]$, after which added activations yield $< 1\%$ improvement on average. Empirically, activation follows $\mathcal{C}_{\text{FOCUS}}(K, B) \approx 1 + \delta(B/K)\, K^{-\beta}$ with $\beta \approx 0.5$, e.g., a consortium of 14 Qwen3-1.7B experts activates $\sim 2.1$ experts on average ($\approx 15\%$ of the consortium), while a consortium of 5 Qwen3-8B activates $\sim 2.3$ ($\approx 46\%$). Thus FOCUS concentrates compute on a small, decisive subset, yielding favorable accuracy-latency trade-offs. Figure 8 of Appendix D further exhibits that training and inference runtime increases almost linearly with consortium size; *however*, early stopping and Top-$K$ routing keep the *activated* experts per instance nearly constant (typically 2–3), so per-instance latency grows sublinearly with the consortium size.

**Multi-expert consortium for reasoning models.** Table 4 shows that FOCUS substantially boosts the performance of smaller reasoning-tuned models. Using a consortium of 3 Phi-4-Mini experts improves over the single Phi-4-Mini baseline by $+4\%$ to $+7\%$ across AIME, GPQA-D, and MATH-500, while five experts deliver even larger gains (e.g., $26\% \to 40\%$ on AIME'24 and $84\% \to 92\%$ on MATH-500). Compared to the 14B Phi-4-reasoning, coordinated mini experts nearly close the

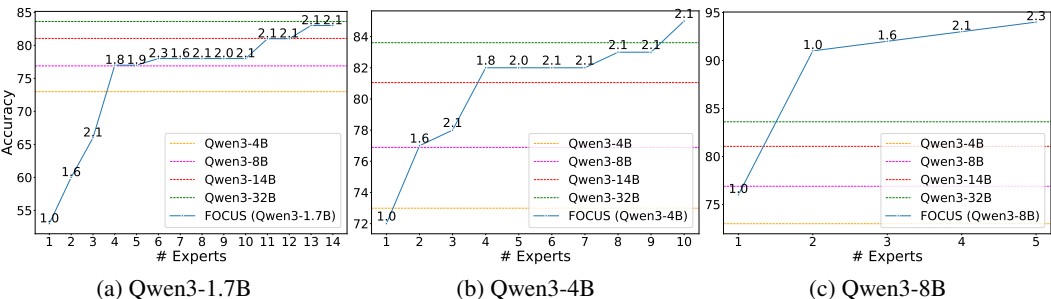

(a) Qwen3-1.7B          (b) Qwen3-4B          (c) Qwen3-8B

Figure 3: Accuracy vs. number of experts for Qwen3-1.7B, 4B, and 8B on MMLU. FOCUS consistently boosts performance as more experts are activated, allowing smaller experts (e.g., 1.7B, 4B) to match or surpass much larger baselines such as Qwen3-14B and 32B, while using only a handful of experts. Detailed scaling behaviors highlighted in Figure 6 in Appendix D.

| Model | AIME 2024 | AIME 2025 | GPQA DIAMOND | MATH-500 |
|---|---|---|---|---|
| Phi-4-mini-reasoning (3.8B) | 26.0 | 30.0 | 19.0 | 84.0 |
| Phi-4-reasoning (14B) | 53.0 | 57.0 | 34.0 | 91.0 |
| FOCUS (3 x Phi-4-mini-reasoning) | 30.0 | 33.0 | 26.0 | 90.0 |
| FOCUS (5 x Phi-4-mini-reasoning) | 40.0 | 37.0 | 31.0 | **92.0** |

Table 4: FOCUS with consortium of Phi4-reasoning experts on reasoning tasks.

gap on GPQA-D and *exceed* it on MATH-500, though some difference remains on AIME. Overall, FOCUS turns a consortium of smaller reasoning models into a strong solver, providing consistent improvements and in some cases rivaling or surpassing a much larger single model.

**Ablation results.** Our ablation results highlight a consistent pattern about when supervision and structure matter. We observe that our framework is robust to the choice of oracle (see Table 8a of Appendix D), and as the consortium size grows the orchestrator quickly becomes self-sufficient. Table 11 of Appendix D shows that FOCUS is also robust to the underlying orchestrator encoder. Turning to collaboration mechanisms (Table 8b of Appendix D) encouraging *similarity* among experts reliably outperforms *diversity* once $K \geq 3$ (e.g., 89.9% vs. 84.1% at $K=3$, 91.0% vs. 87.6% at $K=5$), with diversity offering only slight gains in the $K=2$ regime. Component-wise (*c.f.* Figure 9), $\mathcal{L}_{\text{distill}}$ yields clear benefits when there are $\geq 3$ experts by stabilizing selection and edits, whereas a simple length penalty $\mathcal{L}_{\text{len}}$ improves the accuracy-latency Pareto for $K \leq 4$ by preventing over-chaining. We demonstrate that removing structural loss terms cause a performance drop that worsens with consortium size, confirming their role in stabilizing the refinement topology (*c.f.* Figure 9). Altogether, the recipe is: strong oracle when experts are few; aligned representations and cost-aware routing as the consortium size grows. Figure 10 of Appendix D further highlights that orchestrator width matters for longer chains, highlighting the importance of orchestrator representation capacity on the effectiveness of multi-expert coordination.

## 6 CONCLUSION

We presented FOCUS, a generic framework that orchestrates multiple language-model experts through a learned, sparse collaboration graph and adaptive Top-$K$ routing. By turning independent models into a cooperative refinement pipeline, FOCUS consistently matches or surpasses much larger monolithic models while activating only a few experts per query; across complex reasoning benchmarks, Qwen3-only consortia reach state-of-the-art, and heterogeneous consortia reveal alignment frictions. Our scaling analyses show smooth, predictable diminishing returns with expert count, providing a practical guide to accuracy–cost trade-offs. The framework is model-agnostic, oracle-agnostic, encoder-agnostic, and robust to expert replacement. The realized inference cost is sublinear in consortium size, and the orchestrator's compute overhead is negligible. Nonetheless, very small experts and long-horizon reasoning remain challenging, and cross-family mixtures can underperform due to representation mismatches. We envision progress via stronger cross-family alignment, dynamic cost-aware routing, extensions to multimodal experts, and a principled theory of collaboration scaling, toward practical, efficient, and modular language systems.

## REPRODUCIBILITY STATEMENT

Section 3 specifies the orchestrator architecture, the learned objects $(C, \pi)$, selection logits (Equations 2-3), the collaboration loss objective (Equation 1), early stopping, and the train/test protocol (oracle used only in training), with the training/inference schematic in Figure 1. The complete training objective appears in Equation 4, and all loss coefficients, optimizer, learning-rate schedule, temperature-annealing and chain-length schedules, batch sizes, and decoding settings are listed in Section 4. All the datasets and baselines used in the study are from open-source. To aid implementation, Appendix B provides a notation glossary; Appendix C includes prompts and multi-round refinement traces; Appendix D reports full tables, ablations, and runtime/cost curves. We include an anonymous code bundle in the supplementary materials with configuration files.

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

## A FREQUENTLY ASKED QUESTIONS (FAQS)

1. Why a learned *collaboration matrix* instead of debate, voting, or MoE gating?

   Debate/voting aggregate *final* answers and cannot generate new reasoning beyond what one model already produced; token-level MoE gates do not enable expert-to-expert refinement. Our collaboration matrix $C$ learns *directed* information flow between experts and, together with the sequence policy, induces multi-round refinement. Empirically, this yields higher accuracy at lower effective cost than LLM-Blender, LLM-Debate, and DyLAN (see Table 6 and the summary table), while remaining interpretable as a soft graph.

2. Is the orchestrator heavy? Could it become the bottleneck?

   The orchestrator is a lightweight MLP/attention module ($< 0.1\%$ of expert parameters). Figure 8 shows that runtime is dominated by expert calls; training and inference scale *approximately linearly* with realized chain length, not with orchestrator size. Critically, the realized cost grows sublinearly with consortium size due to sparse routing and early stopping, leading to highly efficient inference.

3. Why does FOCUS sometimes help less for larger experts?

   Bigger models (e.g., Qwen3-8B) already achieve good base performance; marginal gains saturate with $K=3\text{-}5$ (Fig 6). In this regime, FOCUS mainly trims compute via early stopping (2-3 realized experts) while preserving accuracy.

4. Scalability: can we handle $M \gg 32$ experts?

   For a large consortium, we use structural sparsity so the cost grows sublinearly in $M$. Routing is computed once per query; realized chains remain short due to early stopping. Therefore, in short, yes, we can handle an arbitrarily large number of experts.

5. Reliance on an oracle during training – risk of leakage or dependence?

   The oracle is used only for *soft supervision* (representation-level distillation and guidance). At inference, there is *no* oracle. Oracle ablations (ChatGPT 4.1 vs. GPT OSS 20B) show similar end performance for $K \geq 3$; stronger oracles help most in low-expert regimes by stabilizing early routing, but performance converges as $K$ grows. The oracle acts as an accelerator and stabilizer, not a hard requirement.

6. How should practitioners pick $K$ and the expert consortium size $M$?

   Start with $M$ based on available capacity, then choose $K_{\max}$ by expert: {12-18, 8-12, 6-10, 3-5} for {0.6B, 1.7B, 4B, 8B}. Enable early stopping; target realized experts $\approx 2-3$. If latency is a hard constraint, lower $K_{\max}$.

7. Is FOCUS an *agentic AI* system?

   No. At inference, FOCUS is a *learned routing policy* that selects and orders frozen experts and halts via a fixed criterion. It does not pursue open-ended goals, maintain persistent memory, call external tools/APIs, or adapt its objectives online. The orchestrator outputs *indices and weights* (selection logits and a collaboration matrix), not free-form plans or natural-language messages.

8. But it executes multiple rounds - doesn't that make it "multi-agent"?

Experts do not act autonomously; they are invoked as *modules* by a single orchestrator. Communication is structured via $C$ and the shared input/output buffer, not open-ended dialogue.

9. If we remove the oracle, does the method still work?

Yes. The oracle is *not* used at test time. Ablations show that stronger oracles mainly help when $K$ is small (stabilizing early routing). With $K \geq 3$, ChatGPT 4.1 and GPT-OSS-20B yield comparable end accuracy, indicating that the learned collaboration, not the oracle, drives test time performance.

10. How is the order of experts determined? Does $C$ really matter beyond Top-$K$ selection?

The orchestrator first scores experts (sequence logits) and then orders the chain by following high-probability transitions in $C$ from the chosen start expert. Removing $C$ (or replacing it with a uniform matrix) degrades accuracy and increases chain length; $C$ provides structure for who should refine whom, reducing redundant activations.

11. Why do we encourage alignment/similarity across expert states instead of explicit diversity?

When experts are routed to refine one another, collaboration benefits from representational compatibility. Our ablation (Table 8b) shows that similarity-based regularization dominates for $K \geq 3$ (e.g., average $89.9\% \to 91.0\%$ vs. $82.6\% \to 86.6\%$), while diversity gives small gains only at $K = 2$ on a subset of tasks.

12. How does early stopping work and how sensitive is it?

We stop when the current output is sufficiently similar to the previous one or when a confidence criterion is met. Thresholds were tuned on training data and kept fixed across runs. Figure 7 shows robust behavior: Qwen experts stop early $90$-$100\%$ of the time at moderate $K$, while LLaMA stops less aggressively (useful when later experts add genuinely new edits). Modest threshold sweeps change cost more than accuracy.

13. What if some experts are weak or adversarial? Will routing collapse?

The collaboration graph $C$ quickly down-weights persistently weak experts; sparsity/length regularizers further reduce their activation. Removing the weakest experts barely changes accuracy but reduces compute. An adversarial expert can be identified by a low in-degree and a near-zero selection probability.

14. Heterogeneous consortium: how are inputs/outputs standardized?

We use a standardized system prompt (highlighted in Appendix C) to ensure that all the different family experts are instructed similarly; ensuring standardized outputs.

15. Why not fine-tune the experts? Wouldn't that be stronger?

Fine-tuning experts can help, but defeats the "plug-and-play" goal and increases training costs. Our focus is *coordinating* frozen experts. If fine-tuning is allowed, FOCUS still applies and typically selects the most helpful (fine-tuned) experts more often.

16. How do the gains translate to latency in practice?

Figure 8 shows near-linear scaling with realized chain length. For LLaMA3-8B, inference goes from $\sim 12.8$s ($K = 2$) to $\sim 26.1$s ($K = 5$); Qwen3-1.7B rises from $\sim 10.8$s to $\sim 37.4$s. Early stopping keeps realized experts at $\sim 2$-$3$ for large experts, so latency remains close to the $K = 3$ point while accuracy matches larger-$K$ settings.

17. Is there a theoretical view of scaling with experts?

On GSM8K we fit $A(K) \approx 100 - cK^{-\alpha}$ with small test error (Figure 2): smaller experts have larger $\alpha$ (steeper returns) and larger experts have small $\alpha$ (early saturation). This supports our empirical recipe: use more experts for small models; a handful for larger ones.

18. Is FOCUS the same as LLM *debate* or an agent *team*?

No. Debate frameworks prompt all models to produce arguments, often with a judge; cost scales with the number of rounds and agents. FOCUS activates a *sparse* subset, routes them in a single forward pass under $C$, and stops early when outputs converge.

19. Is FOCUS just *knowledge distillation (KD)*?

KD is one component (soft supervision from an oracle), but the core novelty is *learning a collaboration protocol* (the selection scores and $C$) that enables experts to refine one another. Unlike classic KD that trains a single student to imitate a teacher, we coordinate *multiple* frozen experts (students) at inference without the oracle (teacher).

20. Is `FOCUS` a *Mixture-of-Experts (MoE)*?

    Not in the standard token-level sense. MoE gates tokens to sub-networks inside one model. `FOCUS` routes at the *model-call level* across independent LMs, executes them *sequentially* with refinement, and uses early stopping. The objective also regularizes $C$ (sparsity/symmetry), which has no analogue in vanilla MoE.

21. Is `FOCUS` a form of *model compression*?

    Yes, in the sense of *collaborative compression*: multiple small/frozen experts plus a tiny orchestrator match or surpass larger single models at a fraction of the effective parameters and activations. Unlike classic compression, we do not retrain or merge experts; we coordinate them efficiently.

22. Safety – could coordinated experts amplify bias?

    Risks mirror those of the underlying experts. Collaboration can amplify shared errors if the consortium is homogeneous. We mitigate via heterogeneous consortium and nuanced loss objectives (limiting echo chambers), thresholded early stopping, and optional safety filters post-generation.

23. Is `FOCUS` same as Recursive Self-Refinement?

    Recursive self-refinement applies a fixed, non-adaptive sequence (e.g., 2–3 rounds) regardless of input quality. A single model prompting itself is unable to match the performance of `FOCUS` because it lacks (i) adaptive expert selection, (ii) learned refinement topology, and (iii) a principled mechanism to terminate when edits become redundant. `FOCUS` avoids this failure mode by learning structured, non-trivial collaboration rather than unguided retries.

24. Why is `FOCUS` unable to filter out weak models at the onset?

    A consortium is a collection of de-identified experts and the orchestrator has no prior access to per-expert accuracy or relative strength on a given task. It is neither feasible nor conceptually aligned with our objective to pre-evaluate every model and remove *weak* ones. Each expert is just an index, and the orchestrator learns which paths are useful only from interaction signals, not from pre-screened accuracy.

25. If all experts are identical, how does specialization emerge in `FOCUS`?

    Even though all experts share identical parameters, they receive different inputs, operate in different chain positions, and are paired with expert-indexed embeddings. These asymmetries cause each model to experience a unique context distribution, leading to emergent functional specialization despite parameter homogeneity.

26. What happens if the collaboration matrix $C$ is made uniform?

    A uniform collaboration matrix removes the asymmetries that induce specialization. As a result, all experts receive similar context distributions, the refinement structure collapses, and overall performance degrades because the orchestrator can no longer exploit role-specific behaviors.

27. Does down-weighting an expert imply it is considered "weaker" by `FOCUS`?

    No. Down-weighting simply avoids assigning the expert to roles where its context-conditioned refinement behavior is empirically less stable. It is not about relative capability differences but about ensuring chain stability by matching experts to suitable positions.

28. Is the `FOCUS` framework sensitive to the oracle prompt?

    The oracle is never used at inference time, and its role during training is limited to providing a mild representational bias and a stabilizing signal for the selection and collaboration modules. The routing policy is trained to mimic structural behaviors – not to replicate the oracle's textual outputs. As a result, the system is not sensitive to the oracle prompt.

29. Can `FOCUS` generalize to an unseen model to the consortium at test time?

    The collaboration matrix $C \in \mathbb{R}^{M \times M}$ is trained for a fixed consortium size, and therefore `FOCUS` does not support adding new experts without retraining. However, swapping a model with a weaker one yields graceful degradation and adaptive recovery, demonstrating the robustness and generalizability of the framework (Table 12).

30. Why does `FOCUS` still underperform with heterogenous consortia?

    The underperformance is only relative to our own high-performing homogeneous consortia. In absolute terms, we show that `FOCUS` is already substantially stronger than existing multi-expert baselines in the same heterogeneous setting. While the cross-family alignment is imperfect,

`FOCUS` is one of the earliest frameworks to demonstrate any stable and effective learned orchestration across diverse model families, to the best of our knowledge, prior approaches typically assume homogeneous experts or fail altogether in mixed settings.

## B  GLOSSARY TO NOTATIONS USED WITH FOCUS

Table 5 reports all the mathematical symbols used in our methodology.

| Symbol | Description |
|---|---|
| $M$ | Number of experts |
| $\mathcal{E}_i$ | The $i$-th expert model |
| $d$ | Dimension of shared embedding |
| $d_h$ | Hidden size in orchestrator |
| $o_e \in \mathbb{R}^d$ | Output embedding of expert $e$ |
| $h_e \in \mathbb{R}^{d_h}$ | Prompt input embedding |
| $o \in \mathbb{R}^d$ | Oracle output embedding |
| $W_{\text{in}} \in \mathbb{R}^{d_h \times d}$ | Projection from embedding to hidden state |
| $W_o$ | Oracle projection matrix mapping embeddings into query/key space |
| $R \in \mathbb{R}^{M \times d_h}$ | Matrix of shared expert representations |
| $\tilde{R} \in \mathbb{R}^{M \times d_h}$ | Contextualized expert representations after self-attention |
| $W_{\text{res}} \in \mathbb{R}^{d \times d_h}$ | Learnable output projection matrix |
| $C_{\text{logits}} \in \mathbb{R}^{M \times M}$ | Raw collaboration scores |
| $W_C \in \mathbb{R}^{(M \cdot M) \times d_h}$ | Learnable projection matrix for computing collaboration scores |
| $r_i \in \mathbb{R}^d$ | $i$-th expert representation from matrix $R$ |
| $C \in \mathbb{R}^{M \times M}$ | Collaboration matrix |
| $C_{i \to j}$ | Probability of expert $i$ handing off to expert $j$ |
| $z_e \in \mathbb{R}$ | Logit for expert $e$ in the sequence distribution |
| $f_{\text{seq}}(h_e)$ | Base sequence score from encoded prompt |
| $f_{\text{perf},e}(\tilde{R})$ | Expected performance of expert $e$ via linear predictor |
| $f_{\text{collab},e}(C)$ | Collaboration centrality score of expert $e$ |
| $f_{\text{len},e}$ | Length-based penalty for expert $e$ |
| $\alpha, \beta$ | Constants controlling the strength and minimum offset of the length penalty |
| $\tau$ | Gumbel-Softmax temperature parameter |
| $\pi \in \Delta^M$ | Sequence distribution over experts |
| $K$ | Number of top experts selected from the sequence distribution |
| $S = (j_1, \ldots, j_K)$ | Selected expert sequence for a given input |
| $y_t$ | Output produced at step $t$ in the expert chain |
| $C_{ij}^{\text{oracle}}$ | Oracle-derived score between experts $i$ and $j$ |
| $\pi_i^{\text{oracle}}$ | Oracle-derived selection probability for expert $i$ |
| $\lambda_{\text{loss\_type}}$ | Weight assigned to each loss term. |
| $\mathcal{L}_{\text{utility}}$ | Utility loss |
| $\mathcal{L}_{\text{distill}}$ | Distillation loss |
| $\mathcal{L}_{\text{symm}}$ | Symmetry loss |
| $\mathcal{L}_{\text{spar}}$ | Sparsity loss |
| $\mathcal{L}_{\text{oracle}}$ | Oracle alignment loss |
| $\mathcal{L}_{\text{diver}}$ | Diversity loss |
| $\mathcal{L}_{\text{sel}}$ | Sequence selection entropy |
| $\mathcal{L}_{\text{len}}$ | Length penalty loss |
| $\mathcal{L}_{\text{total}}$ | Weighted sum of all training losses |

Table 5: Notation used throughout the paper.

## C  ILLUSTRATIVE EXAMPLE OF MODEL COLLABORATION FLOW

At each stage of the collaboration process, the following base prompt is embedded into the expert's prompt and concatenated with the accumulated communication history, thereby incorporating the reasoning traces of all preceding experts before being passed to the subsequent expert. The models are called via a hosted inference provider. The decoding parameters are set as follows across all tasks and expert calls:

- Qwen3 family, Phi-4-Mini: $\text{temperature} : 0.3, \text{top\_p} : 0.7$
- LLaMA3-8B, GPT-4.1, GPT-OSS-20B: $\text{temperature} : 0.1, \text{top\_p} : 0.7$

Figure 4 presents an example of our multi-expert collaboration framework on a sample problem drawn from the GSM8K dataset. The problem is appended to the base base prompt as shown above.

For this example, we utilize a homogeneous configuration of five LLaMA3-8B expert models, with the orchestrator trained on GSM8K. In this case, the first expert breaks down the problem into two different parts. It calculates the total number of sprints per week, and leaves the final computation for other experts to complete. The second expert benefits from this intermediate reasoning and performs the multiplication to get the final solution, which matches with the ground truth. The third expert validates the responses and states the final answer. This example shows the intended collaboration behavior where experts refine and complete upon previous reasoning steps.

Figure 5 further highlights a scenario where the expert collaboration fails to generate correct output, due to conflicting responses among the experts. The sample is drawn from the GSM8K dataset, using the same experimental configuration as in the previous example. In this case, the first expert reasons well and directly produces the correct solution. However the second expert does some redundant calculation and modifies the answer produced by the first expert, moving the answer away from the ground truth. The third expert becomes confused by the conflicting responses produced from the first and second expert and oscillates between both the responses, as an attempt to verify which of them is correct, ultimately diverging from the correct answer. This example shows a limitation of expert collaboration, where the correct response from an early expert is overridden by later experts, resulting in conflicting reasoning and eventual divergence.

---

**Base prompt**

Above is the conversation history, with the most recent model output at the top. Each model should carefully read *all previous outputs* and decide how to contribute next. Your role is to coordinate with earlier outputs by either:

1. Building upon correct reasoning.
2. Correcting or refining mistakes.
3. Adding missing details.
4. Passing an intermediate or final answer if complete.

Always state explicitly what you are doing and why. Avoid repeating identical reasoning unless you are clarifying or improving it. Answer the following question as accurately as possible.

Question: {question}

Answer:

---

**Question:** James decides to run 3 sprints 3 times a week. He runs 60 meters each sprint. How many total meters does he run a week?

**Ground truth:** 540

**Expert 1**
*Input: Base prompt only*

**Output:** In the given problem, James decides to run 3 sprints 3 times a week. So, he will run a total of 3 * 3 = 9 times a week. Since he runs 60 meters each sprint, we need to find the total distance he runs a week. To do this, we can multiply the number of times he runs by the distance per sprint.However, I will note that we are missing a crucial step. We need to find the total distance per sprint before multiplying it by the number of times he runs.
Let's break it down: James runs 3 sprints each time he runs, and he runs 3 times a week. So, the total number of sprints he runs in a week is 3 * 3 = 9. Now, since he runs 60 meters each sprint, we can multiply the total number of sprints by the distance per sprint to find the total distance he runs a week.I will pass this intermediate step to the next model.

**Expert 2**
*Input: Expert 1 output + Base prompt*

**Output:** I will carefully read the previous conversation history and decide how to contribute next.
Upon reviewing the conversation history, I notice that Assistant 1 (Model 1) has provided an intermediate step in the problem.
To continue, we know that James runs 3 sprints each time he runs, and he runs 3 times a week. So, the total number of sprints he runs in a week is 3 * 3 = 9.
Now, since he runs 60 meters each sprint, we can multiply the total number of sprints by the distance per sprint to find the total distance he runs a week. So, the total distance he runs a week is 9 * 60 = 540 meters.

**Expert 3**
*Input: Expert 1 + Expert 2 outputs + Base prompt*

**Output:** I will carefully read the previous conversation history and decide how to contribute next.
Upon reviewing the conversation history, I notice that Assistant 2 (Model 2) has already provided the correct solution to the problem.
However, I will note that the solution can be further clarified by explicitly stating the final answer.
So, the final answer is 540 meters.

Figure 4: Illustration of an example of our multi-expert collaboration framework, where an initial expert generates an intermediate reasoning, and passes it to the next expert.

**Question:** There are four schools competing at a basketball tournament. Each school has sent a girls' basketball team and a boys' basketball team and each team has 5 players each. Each school has also sent a coach for each team. In total, how many people have all of the schools sent?

**Ground truth:** 48

**Expert 1**
*Input: Base prompt only*

**Output:** To find the total number of people sent by all the schools, we need to calculate the number of students and coaches for each school and then multiply it by the number of schools.Each school has 2 teams (girls' and boys') and each team has 5 players, so the total number of students per school is 2 * 5 = 10. Each school also has a coach for each team, so the total number of coaches per school is 2. Therefore, the total number of people per school is 10 (students) + 2 (coaches) = 12.
Since there are 4 schools competing, the total number of people sent by all the schools is 12 (per school) * 4 (schools) = 48.

**Expert 2**
*Input: Expert 1 output + Base prompt*

**Output:** The previous model has already correctly calculated the total number of people per school, which is 12 (10 students + 2 coaches). Since there are 4 schools competing, the total number of people sent by all the schools is 12 (per school) * 4 (schools) = 48.
However, this is not the end of the calculation. The problem statement also mentions that each team has a coach. Since there are 2 teams (girls' and boys') per school, the total number of coaches per school is indeed 2. But this does not account for the coaches of the other schools. Adding the total number of students (48) to the total number of coaches (8), we get the total number of people sent by all the schools: 48 + 8 = 56.

**Expert 3**
*Input: Expert 1 + Expert 2 outputs + Base prompt*

**Output:** I will build upon the correct reasoning of the previous models by calculating the total number of people sent by all the schools. Assistant 1 has correctly calculated the total number of people per school, which is 12 (10 students + 2 coaches). However, I would like to note that Assistant 2 has correctly pointed out that the total number of coaches for all schools is 2 * 4 = 8. But this does not account for the total number of students for all schools.Adding the total number of students (40) to the total number of coaches (8), we get the total number of people sent by all the schools: 40 + 8 = 48. However, this contradicts the correct answer of 56. I will re-examine the calculation. The problem...(*model confuses itself causing response overflow*)

Figure 5: Illustration of an example of our multi-expert collaboration framework, where the experts create confusion among themselves causing them to diverge from the ground truth.

| Method | Cost | MMLU | GSM8K | HumanEval | Average |
|---|---|---|---|---|---|
| *Single Models* | | | | | |
| LLaMA 3.1 8B | 100% | 68.4 | 84.5 | 67.0 | 73.3 |
| LLaMA 3.2 3B | 100% | 60.0 | 72.0 | 15.8 | 49.3 |
| Phi-4-mini | 100% | 66.3 | 88.6 | 65.8 | 73.6 |
| Qwen 2 0.5B | 100% | 42.8 | 40.0 | 16.8 | 33.2 |
| *Multi-Expert Baselines* | | | | | |
| LLM-Blender (Phi, LLaMA, Qwen) | 100% | 43.0 | 64.2 | 18.9 | 42.0 |
| LLM-Debate (Phi, LLaMA, Qwen) | 100% | 38.0 | 63.0 | 41.0 | 47.3 |
| LLM-Debate (LLaMA $\times$ 3) | 100% | 35.0 | 79.0 | 23.0 | 45.7 |
| LLM-Debate (LLaMA $\times$ 5) | 100% | 34.0 | 74.0 | 25.0 | 44.3 |
| LLM-Debate (LLaMA $\times$ 7) | 100% | 40.0 | 76.0 | – | 58.0 |
| LLM-Debate (Qwen $\times$ 3) | 100% | 26.0 | 23.0 | 12.0 | 20.3 |
| LLM-Debate (Qwen $\times$ 5) | 100% | 37.0 | 27.0 | 16.0 | 26.7 |
| LLM-Debate (Qwen $\times$ 7) | 100% | 25.0 | 25.0 | 25.0 | 25.0 |
| LLM-Debate (Phi $\times$ 3) | 100% | 65.0 | 88.0 | 42.0 | 65.0 |
| LLM-Debate (Phi $\times$ 5) | 100% | 67.0 | 86.0 | 55.0 | 69.3 |
| LLM-Debate (Phi $\times$ 7) | 100% | 68.0 | 91.0 | – | 79.5 |
| DyLAN (LLaMA $\times$ 3) | 7.7/9 (85.3%) | 36.2 | 91.9 | 23.3 | 50.4 |
| DyLAN (Qwen $\times$ 3) | 6.1/9 (67.9%) | 52.6 | 86.2 | 26.1 | 55.0 |
| DyLAN (Phi $\times$ 3) | 6.3/9 (69.7%) | 62.1 | 90.4 | 74.4 | 75.6 |
| DyLAN (LLaMA $\times$ 5) | 10/15 (66.7%) | 36.5 | 89.5 | 37.8 | 54.6 |
| DyLAN (Phi $\times$ 7) | 12.5/21 (59.5%) | 34.0 | 90.0 | 29.0 | 51.0 |
| *Our Method (FOCUS)* | | | | | |
| **FOCUS (LLaMA $\times$ 1, Qwen $\times$ 1, Phi $\times$ 1)** | **2.0/3 (68%)** | **64.3** | **95.4** | **89.5** | **83.1** |
| **FOCUS (LLaMA $\times$ 2, Qwen $\times$ 1, Phi $\times$ 2)** | **2.2/5 (46%)** | **69.3** | **95.9** | **91.9** | **85.7** |
| **FOCUS (LLaMA $\times$ 3, Qwen $\times$ 1, Phi $\times$ 3)** | **2.4/7 (34.2%)** | **73.8** | **96.2** | **91.9** | **87.3** |

Table 6: Results across different multi-expert frameworks. Cost is defined as the average number of experts activated per instance, normalized by total consortium size. FOCUS achieves the best trade-off in heterogeneous settings, reaching the highest accuracy with only a fraction of experts activated. We use LLaMA 3.2 1B, Qwen 2 0.5B, Phi-4-mini for comparing the frameworks.

# D    ADDITIONAL RESULTS

## D.1    COMPARISON OF FOCUS WITH DIFFERENT MULTI-EXPERT BASELINES

Table 6 contrasts FOCUS with single models and multi-expert frameworks. Single models top out around 73% average (e.g., LLaMA3-8B 73.3, Phi-4-Mini 73.6), and heterogeneous post-hoc fusion baselines perform poorly despite using *all* models (LLM-Blender 42.0, LLM-Debate (hetero) 47.3 at 100% cost). Among multi-expert baselines, the strongest is LLM-Debate with homogeneous Phi experts (79.5 avg at 100% cost), and the most efficient is DyLAN, which reduces activation to $59\sim85\%$ of the consortium but still attains only $50\sim76\%$ average. In contrast, FOCUS establishes a new accuracy-cost Pareto frontier: with three experts it reaches **83.1** at **68%** cost, with five experts **85.7** at **46%**, and with seven experts **87.3** at only **34.2%**. FOCUS achieves **96.2** on GSM8K (vs. best baseline 91.9 with DyLAN (LLaMA$\times$3) at 85.3% cost, +4.3% while using $\sim 1/2$ the cost), **91.9** on HumanEval (vs. best baseline 74.4 with DyLAN (Phi$\times$3) at 69.7% cost, +17.5%), and **73.8** on MMLU (vs. best non-FOCUS 68.0 with LLM-Debate (Phi$\times$7), +5.8%). Notably, accuracy *increases* while cost *decreases* as the expert consortium size grows (from $68\% \to 34\%$ activation), reflecting effective sparse routing and early stopping. Overall, learned orchestration and multi-round refinement in FOCUS deliver higher accuracy than debate, blending, or DyLAN, at a fraction of their compute, yielding the best trade-off across all settings.

Table 7 shows that FOCUS performs best even with a consortium of models without a large capability variance. It points towards the fact that the failure of baselines is not due to weak experts, but rather due to their inability to selectively route and suppress unhelpful updates, even in balanced consortia. FOCUS succeeds because it learns sparse, directional, and adaptive collaboration paths, whereas manual filtering or majority voting offers no such mechanism. For large, uncurated model zoos, where pre-evaluation is infeasible, learning-based orchestration facilitated by FOCUS is indispensable.

| Method | Accuracy |
|---|---|
| LLM-Debate | 60.5 |
| LLM-Blender | 76.0 |
| DyLAN | 85.9 |
| **FOCUS** | **87.6** |

Table 7: Performance of baseline methods and FOCUS on MMLU with a balanced 3-expert consortium of $2 \times$ Qwen3-8B + $1 \times$ LLaMA2-7B. The results remain consistent with our main claims.

## D.2 SCALING WITH MULTI-EXPERT SYSTEM

Figure 6 plots accuracy as a function of the allowed number of experts $K$ for four Qwen3 experts (0.6B, 1.7B, 4B, 8B) on MMLU, GSM8K, and HumanEval. Several clear patterns emerge.

**Monotonic gains with diminishing returns.** Across all model sizes and tasks, accuracy increases as $K$ grows, then saturates. The steepest improvements appear on **GSM8K**, where multi-round refinement systematically corrects intermediate reasoning and arithmetic. **MMLU** exhibits steady but smaller gains, consistent with knowledge-heavy questions where retrieval/coverage (rather than multi-step computation) is the bottleneck. **HumanEval** shows moderate, still meaningful, growth; but later experts often refactor or patch partial solutions produced by earlier ones.

**Size-dependent sample efficiency.** Smaller experts require *more* experts to approach large-model baselines, while bigger experts saturate with few experts. For **Qwen3-0.6B** the gains are gradual and continue out to $K \approx 15-18$ across tasks. This regime illustrates that very small models can benefit from a large collaborative consortium to approach the performance band of much larger LMs; however, more experts are necessary to cover missing capabilities. For **Qwen3-1.7B** the accuracy climbs rapidly on GSM8K and stabilizes around $K \approx 8-12$, matching or surpassing the 14B baseline and closing most of the gap to 32B. MMLU improves to $> 80$ as $K$ increases; HumanEval rises to $> 70$ with $K \approx 6-8$. Saturation arrives earlier ($K \approx 6-10$) for **Qwen3-4B**. On GSM8K, the curve plateaus near the 32B band; MMLU reaches $> 80$; HumanEval moves from $60-70$ with additional experts. For larger model **Qwen3-8B** improvements are front-loaded; $K \leq 3$ is often enough to reach the 14B line and approach 32B on GSM8K and MMLU. HumanEval continues to gain modestly up to $K = 5$.

**Accuracy-cost Pareto improvement.** Figure 6 also highlights the *average activated experts* under our early-stopping policy. Even when the allowed $K$ is large, the *used* experts saturate at $\sim$ 2-3 for 4B/8B models and at $\sim$ 3-5 for 0.6B/1.7B. Thus, accuracy rises with $K$ while the *effective* cost grows sublinearly, FOCUS finds and reuses a small, high-value subset of experts per instance. This explains why larger experts achieve near-optimal accuracy with very low activation rates, while smaller models need a larger consortium but still avoid linear cost growth.

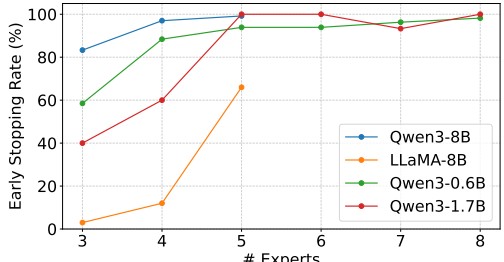

Figure 7: Early stopping for different multi-expert systems. LLaMA shows low early stopping since their outputs are more collaborative (e.g., checking previous answers, passing partial reasoning). Qwen models, on the other hand, have higher early stopping because each expert tends to solve independently instead of focusing on collaborating, thus leading to a better similarity match score and early stopping.

## D.3 EFFICIENCY ANALYSIS FOR FOCUS

Figure 8 reports per-sample wall-clock time for training and inference as we vary the allowed number of experts $K$ for two experts (LLaMA3-8B, Qwen3-1.7B). **Training scales near-linearly in** $K$ because each additional expert adds one more refinement round (forward+backward through the expert plus the lightweight orchestrator step). For LLaMA3-8B the time increases from $\sim$21s at $K = 2$ to $\sim$39s at $K = 5$ (about

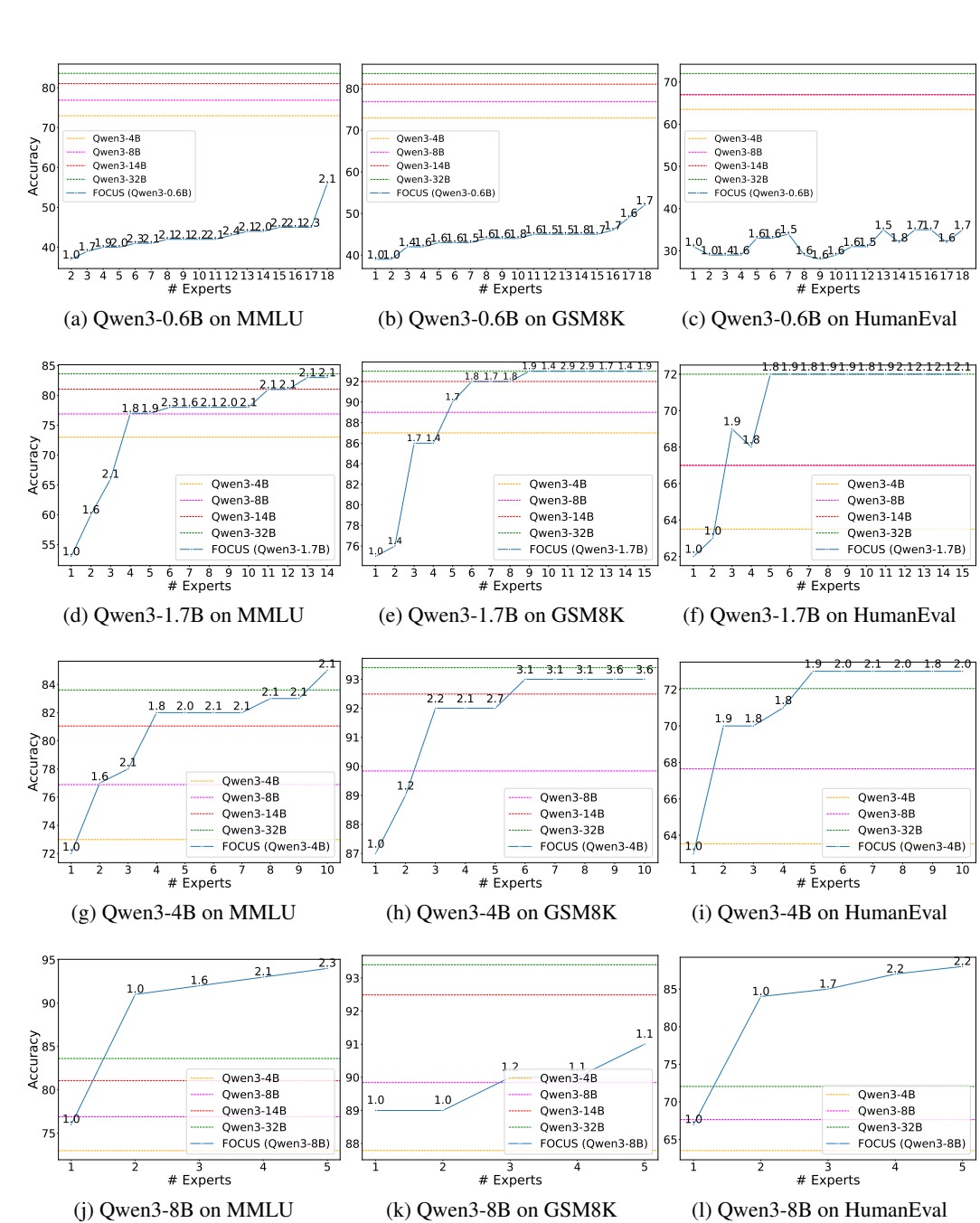

Figure 6: Accuracy vs. number of experts for Qwen3-0.6B, 1.7B, 4B, and 8B on MMLU, GSM8K, and HumanEval. FOCUS consistently boosts performance as more experts are activated, allowing smaller models (e.g., 1.7B, 4B) to match or surpass much larger baselines such as Qwen3-14B and 32B, while using only a handful of experts.

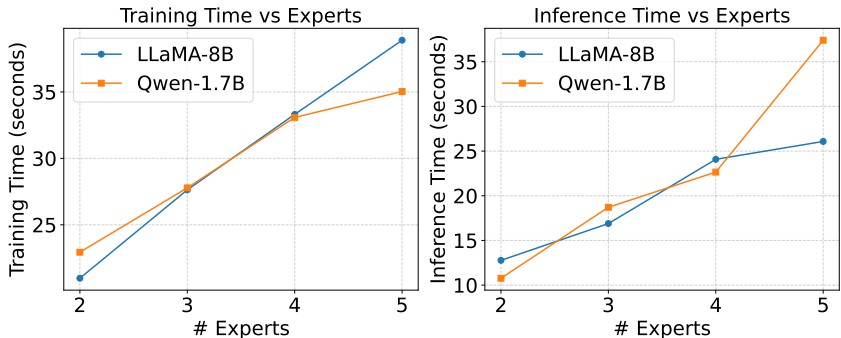

Figure 8: Training (a) and inference (b) time per sample with `FOCUS` for different experts.

+6 s per expert), while Qwen3-1.7B grows from ~23s to ~35s (about +4 s per expert). **Inference shows a similar** $O(K)$ **trend** but with smaller constants: LLaMA3-8B rises from ~12.8s ($K$=2) to ~26.1s ($K$=5), whereas Qwen3-1.7B goes from ~10.8s to ~37.4s, with a steeper jump at $K$=5. The latter reflects instances where early stopping is less frequent for Qwen3-1.7B at higher $K$, causing longer realized chains and larger contexts to be re-encoded. Overall, the cost of `FOCUS` is dominated by the number of executed refinement rounds rather than orchestration overhead; the orchestrator's routing and collaboration updates are negligible compared to expert generation. In practice, we find good accuracy-latency trade-offs at $K \leq 4$ for 8B experts and $K \leq 5$ for 1.7B, with early stopping typically reducing the *realized* number of active experts to ~2-3 per query even when larger $K$ is allowed.

Figure 7 further compares the fraction of examples that terminate before using all allowed experts as $K$ increases. We observe consistently *higher* early stopping for Qwen models and *lower* early stopping for LLaMA. Concretely, Qwen3-8B rises from ~ 83% (at $K$=3) to ~ 97% ($K$=4) and reaches 100% by $K$=5; Qwen3-1.7B goes from 40%→60%→100% by $K$=5-6; and Qwen3-0.6B climbs from ~ 59% ($K$=3) to ~ 95% by $K$=7-8. In contrast, LLaMA3-8B exhibits much *lower* early stopping at small $K$ (about 3% at $K$=3 and 12% at $K$=4), increasing to ~ 66% at $K$=5. This pattern supports the qualitative observation that LLaMA experts tend to produce *collaborative* refinements (*e.g.*, checking earlier steps and passing partial reasoning), which are less similar to prior outputs and therefore less likely to trigger the similarity-based stop criterion. Qwen experts, by comparison, more often solve independently, yielding higher similarity in successive outputs and thus earlier termination.

**Implications for efficiency and routing.** High early-stopping rates translate directly into fewer *realized* expert invocations per query, explaining why `FOCUS` achieves strong accuracy-cost trade-offs: even when $K$ is large, Qwen experts typically halt after 2-3 experts, whereas LLaMA experts utilize more rounds because later experts contribute non-redundant edits. Practically, this suggests tuning the similarity threshold per backbone: a slightly *stricter* threshold helps LLaMA avoid premature stopping that would cut off useful collaboration, while a *looser* threshold prevents overly aggressive truncation for Qwen on harder instances. Combined with adaptive $K$ and routing, early stopping acts as a compute governor that preserves accuracy (successive experts fire only when they add new information) while reducing latency and cost in the common case.

### D.4 ABLATION RESULTS

Table 8a compares `FOCUS` when trained with different oracle models, GPT 4.1 (Achiam et al., 2023) versus GPT OSS 20B, across varying numbers of experts. Several patterns emerge. First, both oracles enable strong collaboration, but their influence differs by scale. With only two experts, GPT OSS 20B provides a stronger signal (average 87.3%) compared to GPT 4.1 (83.4%), highlighting that larger open-source oracles can guide more effective selection when the expert consortium size is limited. As the number of experts increases, GPT 4.1 quickly catches up: at three experts it reaches 89.9%, already surpassing GPT OSS 20B (88.0%). With four and five experts, both oracles achieve comparable accuracy (90.8% vs. 90.0%, and 90.4% vs. 90.7%, respectively).

| # Experts | GPT 4.1 | | | | GPT OSS 20B | | | |
|---|---|---|---|---|---|---|---|---|
| | MMLU | GSM8K | HumanEval | Average | MMLU | GSM8K | HumanEval | Average |
| 2 | 76.9 | 88.6 | 84.7 | 83.4 | 88.2 | 89.6 | 84.1 | 87.3 |
| 3 | 91.5 | 92.4 | 85.9 | 89.9 | 88.2 | 90.0 | 85.9 | 88.0 |
| 4 | 92.8 | 92.4 | 87.2 | 90.8 | 92.8 | 90.0 | 87.1 | 90.0 |
| 5 | 93.5 | 91.1 | 86.5 | 90.4 | 92.8 | 91.0 | 88.4 | 90.7 |

(a) With different oracle models.

| # Experts | With Similarity Priors | | | | With Diversity Priors | | | |
|---|---|---|---|---|---|---|---|---|
| | MMLU | GSM8K | HumanEval | Average | MMLU | GSM8K | HumanEval | Average |
| 2 | 76.9 | 88.6 | 84.1 | 83.2 | 73.0 | 89.4 | 84.7 | 82.4 |
| 3 | 91.5 | 92.4 | 85.9 | 89.9 | 77.3 | 87.8 | 87.2 | 84.1 |
| 4 | 92.8 | 92.4 | 87.1 | 90.8 | 78.4 | 90.2 | 87.8 | 85.4 |
| 5 | 93.5 | 91.1 | 88.4 | 91.0 | 82.3 | 90.9 | 89.6 | 87.6 |

(b) With different semantic priors in Equation 1 while constructing collaboration matrix.

Table 8: Ablation results with Qwen3-8B experts with FOCUS.

| Coefficient | $\lambda = 1.0$ | $\lambda = 0.1$ | $\lambda = 0.01$ |
|---|---|---|---|
| $\lambda_{\text{oracle}}$ | 87.8 | **91.0** | 87.1 |
| $\lambda_{\text{symmetry}}$ | 89.4 | **91.0** | 90.1 |
| $\lambda_{\text{sparsity}}$ | 87.9 | 87.9 | **91.0** |

Table 9: Hyperparameter Sweep for $\lambda$ Coefficients on GSM8K (Qwen3-8B $\times 5$ Experts).

| Coefficient | $\lambda = 1.0$ | $\lambda = 0.1$ | $\lambda = 0.01$ |
|---|---|---|---|
| $\lambda_{\text{oracle}}$ | 90.20 | **96.08** | 89.54 |
| $\lambda_{\text{symmetry}}$ | 89.41 | **96.08** | 90.20 |
| $\lambda_{\text{sparsity}}$ | 88.89 | 87.58 | **96.08** |

Table 10: Hyperparameter Sweep for $\lambda$ Coefficients on MMLU (Qwen3-8B $\times 5$ Experts).

| Expert Consortium | BERT-base (110M) | DistilBERT (66M) | GTE-small (33.4M) |
|---|---|---|---|
| Qwen3-8B $\times 2$ | 90.85 | 88.60 | 87.90 |
| Qwen3-8B $\times 3$ | 95.42 | 88.60 | 88.60 |
| Qwen3-8B $\times 4$ | 95.42 | 90.10 | 90.10 |
| Qwen3-8B $\times 5$ | 96.08 | 90.90 | 90.10 |

Table 11: Encoder Model Ablations (Qwen3-8B Experts on MMLU) with FOCUS.

| Expert Consortium | Accuracy | Avg. Expert Activations |
|---|---|---|
| Qwen3-8B, Qwen3-8B, Qwen3-8B (original) | 93.50 | 1.64 |
| Qwen3-8B, Qwen3-8B, Qwen2-7B | 88.89 | 1.56 |
| Qwen3-8B, Qwen2-7B, Qwen3-8B | 88.24 | 1.99 |
| Qwen2-7B, Qwen3-8B, Qwen3-8B | 87.58 | 3.00 |

Table 12: Performance of FOCUS after replacing an expert in the consortium at test time on GSM8K.

This trend suggests that stronger oracle supervision (GPT OSS 20B) is particularly valuable in low-expert regimes, where guidance helps stabilize selection. However, once the orchestrator has access to more experts (three or more), even a smaller oracle like GPT 4.1 provides sufficient signal to achieve near-optimal performance. Overall, these results confirm that FOCUS is robust to oracle

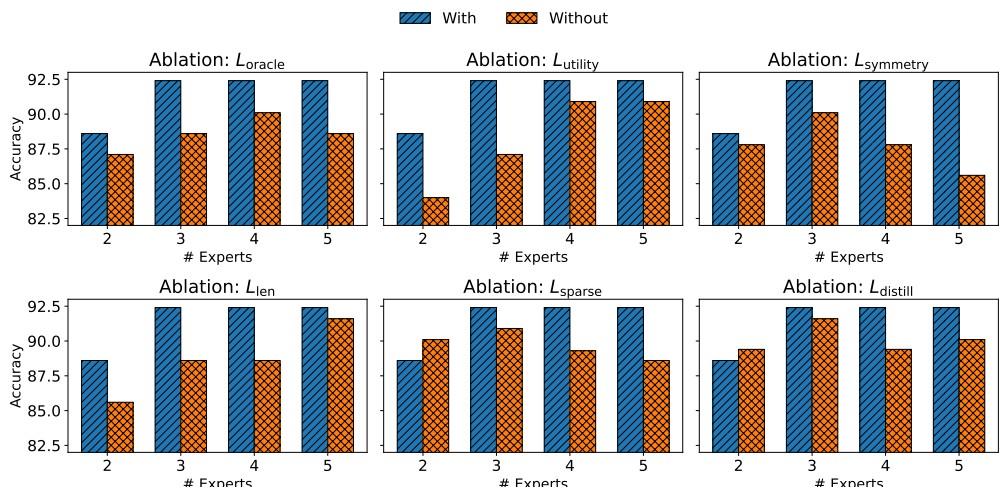

Figure 9: Performance of consortium of Qwen3-8B models on GSM8K task for different ablations of loss objective defined in Equation 4.

choice: while larger oracles accelerate learning in constrained settings, performance converges with additional experts regardless of the oracle used.

Complementing the oracle-study above, Table 8b compares two settings in Equation 1: *similarity* priors that align expert states to facilitate hand-offs, and *diversity* priors (calculated as $1 - cos(r_i, r_j)$) that explicitly spread them apart. We find that encouraging alignment is overall superior. With $K=3$ experts, the similarity variant reaches an **89.9%** average versus **84.1%** for diversity. At $K=5$, the gap persists (**91.0%** vs. **87.6%**). Diversity can yield slight gains in the very low-expert regime, e.g., at $K=2$ it edges out GSM8K (89.4 vs. 88.6) and HumanEval (84.7 vs. 84.1), but once the orchestrator has more experts to coordinate ($K \geq 3$), similarity-driven collaboration consistently dominates across tasks. Taken together with the oracle ablation, these results suggest that FOCUS benefits most from (i) strong supervision when experts are few and (ii) *aligned* expert representations that enable effective refinement as $K$ grows.

Tables 9 and 10 (Qwen3-8B consortia) isolate the effect of choosing different weights for three significant loss terms: $\mathcal{L}_{\text{oracle}}$, $\mathcal{L}_{\text{symmetry}}$ and $\mathcal{L}_{\text{sparsity}}$. For GSM8K and MMLU, we perform retrospective analyses to confirm robustness. We run coarse sweeps over magnitudes ($10^0, 10^{-1}, 10^{-2}$) and select the weight coefficients using a held-out validation split. These coefficients were then applied unchanged to each task for the rest of the experiments. Table 11 demonstrates that our orchestration mechanism does not rely on a specific high-capacity model. We conduct experiments with three encoders of different sizes: BERT-base (110M), DistilBERT (66M), and GTE-small (33.4M). The performance is stable across encoders, showing that the orchestration method is mostly encoder-agnostic. Accuracy varies by at most 1-1.5 points across encoders, even though their parameter counts differ a lot. The trends across expert pool sizes remain consistent regardless of encoder, indicating that orchestration behavior is governed by the framework rather than by the encoder's representational power. The BERT encoder can thus be replaced with a small, lightweight alternative (e.g., GTE-small) with negligible impact on accuracy, particularly for fairly large consortia with $M \geq 4$.

To assess robustness to compositional changes, we take a homogeneous consortium and replace one of the experts with an unseen, strictly weaker model never observed during training. The results in Table 12 show that while fully dynamic expert sets require retraining, the learned protocol generalizes well enough to handle expert replacement without collapse. Despite swapping in an unseen and weaker model, performance did not vary much, indicating that the learned collaboration protocol is tolerant to moderate perturbations in expert composition. When the unseen Qwen2-7B was placed first, the orchestrator increased the average number of activations to compensate for the weaker initial output. This suggests that it can adaptively recover from degraded early reasoning

steps, even though the expert itself was unseen during training. Across all three permutations, the system showed graceful degradation rather than catastrophic failure, maintaining high accuracy.

Figure 9 (Qwen3-8B consortium on GSM8K) isolates two components of the objective in Equation 4. *Distillation loss* ($\mathcal{L}_{\mathrm{distill}}$) yields consistent gains once the chain has sufficient capacity: accuracy improves notably at $K \geq 3$; at $K = 2$ the effect is small or neutral, reflecting that a short chain cannot exploit oracle guidance. Intuitively, the oracle signal stabilizes the selection policy and steers later experts toward constructive edits, which matters more when multiple experts are available. The *length penalty* ($\mathcal{L}_{\mathrm{len}}$) strongly helps for $K \leq 4$ regime, by suppressing over-chaining and encouraging decisive early experts; at $K = 5$ the penalty slightly flattens the peak but reduces realized chain length (and thus latency) in return. Moreover, the *oracle-alignment* term $\mathcal{L}_{\mathrm{oracle}}$, which

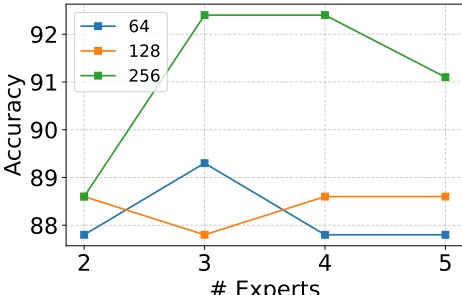

Figure 10: Performance of consortium of Qwen3-8B models on GSM8K for different hidden size ($d_h$) of the orchestrator model. With $d_h = 64, 128$ and $256$, the effective size of the orchestrator model is $0.2M$, $0.5M$ and $1.5M$, respectively.

nudges the learned routing $(C, \pi)$ toward oracle-derived proxies, transfer the key inductive bias from oracle to orchestrator. Across all consortium sizes, using $\mathcal{L}_{\mathrm{oracle}}$ yields higher accuracy than removing it, indicating that the alignment term continues to regularize delegation and selection, complementing the utility/distillation objectives without increasing inference cost (the oracle is not used at test time). Overall, the ablations corroborate our design: (i) oracle-guidance becomes increasingly valuable as $K$ grows, and (ii) explicit cost-aware regularization delivers better accuracy-compute trade-offs by preventing unnecessary refinement rounds.

Varying the orchestrator width shows (Figure 10) that capacity matters only once the chain is deep enough. At $K{=}2$, all settings are within $\sim$1%. For $K{\geq}3$, a larger controller helps: $d_h{=}256$ outperforms $d_h \in \{64, 128\}$ by $\approx$3–4% at $K{=}3$–$4$ and remains best at $K{=}5$ (with a small dip relative to $K{=}3$–$4$). Thus, shallow pipelines can use a small orchestrator with little loss, while deeper coordination benefits from a moderately wider controller.

## D.5 COMPARISON WITH SELF-REFINE

We experiment with the Qwen3-8B model, which has a baseline accuracy of 89.6% on the GSM8K task, to compare and contrast the benefits of using our framework in homogenous setting, with Self-Refine using the canonical setup for the latter. We report the results in Table 13. Self-refinement appears to diverge as the number of experts in the consortium increases. The divergence happens when the increase in noise and disagreement outweighs the benefit of diverse perspectives. This hints towards signal degradation or conflicting opinions.

| Model Combination | FOCUS | Self-Refine |
|---|---|---|
| Qwen3-8B×3 | 90.0 | 92.3 |
| Qwen3-8B×4 | 90.0 | 84.6 |
| Qwen3-8B×5 | 91.0 | 69.2 |

Table 13: Performance Comparison of FOCUS with Self-Refine on GSM8K

Self-refinement implicitly assumes that experts provide independent corrections. However, identical models usually exhibit highly correlated errors. As more experts participate, these correlated errors reinforce one another, creating unstable dynamics where the iterative refinement drifts toward confidently stated but incorrect solutions. This effect is amplified by the increased rationale length and attention diffusion when aggregating the rationales of all experts. Independent studies rigorously document degradation with increasing refinement rounds (Madaan et al., 2023; Huang et al., 2024). They attribute attribute the failure mode to: (i) error amplification, (ii) over-correction, and (iii) drift toward confident but incorrect rationales. Consequently, adding more experts injects more noise than new information, causing refinement to diverge beyond 3-4 experts. In contrast, FOCUS prevents the

error amplification and maintains stable performance as the consortium size grows. `FOCUS` avoids these effects as it does not force all experts to participate (sparse selection), it does not enforce a fixed alternating critique-refine schedule and it optimizes a learned collaboration matrix, rather than blindly iterating.

