# OpenReview forum: "Orchestrated Sparse Consortium of Small Experts Beats Monolithic LLMs"
_ICLR.cc/2026/Conference — Submitted to ICLR 2026_

### Official Review · Reviewer_LnqM · 2025-10-28

**Soundness:** 1
**Presentation:** 2
**Contribution:** 2
**Rating:** 2
**Confidence:** 4

**Summary:**

This paper trains a small orchestrator to manage a pool of individual LLMs. The orchestrator determines which LLM generates the initial response, which LLMs refine the response, and when to output a final answer.

While performance improvements are observed, some of the paper’s claims are questionable. Additionally, the orchestrator’s design, as well as the root causes of the observed improvements, require stronger support and clearer explanation.

**Strengths:**

Downstream accuracy improvements are observed across a diverse set of model pools.

**Weaknesses:**

1. I am curious about the effectiveness of the design. The expert selected first (based on the highest score it receives) is used to generate the initial solution; a high score implies this expert is better suited to address the task initially. However, for refinement, instead of recomputing scores to identify which expert is best suited for refining the solution, the authors reuse the previously computed scores. This means the expert selected for refinement is the second-highest scorer for generating the initial solution. This a choice that seems counterintuitive.

2. In Line 102, the statement “This inverse scaling law justifies the higher effectiveness of FOCUS on smaller language models” is circular. First, you observe that FOCUS performs more effectively on small models. Then, you fit a scaling law to these observed performance patterns. Finally, you claim this fitted law “justifies” that FOCUS works better on small models—using the result of the analysis as evidence for the very phenomenon the analysis was based on.

3. The loss function (Equation 4) includes 8 hyperparameters, making the method overly complex. Tuning such a large number of loss components is highly cumbersome, yet I did not find any ablation studies on L_{sel}, L_{len}, L_{symm}, or L_{spar}. Without these analyses, it is unclear whether such complexity is necessary, nor can we gain insights into the contribution of each component.

4. Table 2 shows that FOCUS is better than LLM-Blender and LLM-Debate. I did not find which models are used for LLM-Blender and LLM-Debate. It is perplexing that LLM-Blender scores only 42% while a single model already achieves 73.6% accuracy. I assume that even a simple voting mechanism should at least match the accuracy of individual models in the pool. Are there many weak models in LLM-Blender's pool? Similar inconsistencies appear in the LLM-Debate results, where the reported accuracy also falls unexpectedly low relative to the single-model baseline (47.3 to 73.6).

5. The method does not train specialized expert models but only trains the orchestrator. For scenarios with multiple homogeneous experts (e.g., two Qwen3-4B models), my understanding is that selecting either of these two Qwen3-4B models is basically the same, as their core capabilities are the same. If this is the case, the "selection" between the two Qwen3-4B reduces to a multi-agent role-playing: their only distinction lies in the roles assigned to the Qwen3-4B models (e.g., refining the initial solution or generating the final answer, which is driven by the prompt).

6. The authors emphasize that the orchestrator is lightweight, with a size of only ~1M parameters. However, the paper also introduces a shared encoder based on BERT. Excluding this encoder when reporting the orchestrator’s size is misleading, as it overstates the method’s orchestration efficiency.

**Questions:**

1. During inference, how is the calling order of selected experts determined?

2. See Weakness 4.

3. Please address the concern raised in weakness 5. For instance, in Table 1b, ~2.3 activated Qwen3-8B models achieve 94.1% accuracy on MMLU. I am curious whether using your prompt, recursively having a single Qwen3-8B model refine its own output ~2.3 times would produce similar accuracy.

4. You claim that small models benefit more from orchestration, but this result is not surprising—it aligns with findings from many RLVR papers, where small models achieve substantially higher pass@K accuracy when rolled out enough times, while large models show minimal pass@K gains as K increases. This pattern is intuitive: small models often make simple, recoverable mistakes, so giving them more opportunities to retry (via refinement or orchestration) allows them to correct errors. In contrast, large models already make fewer mistakes, so additional refinement, retries, or orchestration naturally yield little improvement. Furthermore, given the extremely small number of trainable parameters in your system, I question whether the model can actually learn effective orchestration strategies.

---

> ### Author Response · Authors · 2025-11-18
> **Response to Reviewer LnqM Comments - Part I**
>
> We sincerely appreciate your valuable feedback and further thank you for your comments. We address your concerns and answer your questions in the following response:
>
> ## Ablation
>
> We appreciate the reviewer’s concern regarding the number of loss components and their associated weights. While a full continuous hyperparameter sweep is infeasible within the rebuttal period, we conducted **target-off (binary) ablations** for each loss term on GSM8K across four consortium sizes. These experiments reveal a consistent pattern:
>
> - **Largest contributors:** the *utility loss*, *oracle alignment*, *symmetry*, and the *length penalty* provide the most significant single-term improvements.
> - **Moderate to small contributors:** the *selection entropy* and *distillation* terms offer smaller but stable gains.
> - **Correlated structural terms:** the effects of *symmetry* and *sparsity* covary strongly across consortium sizes (Pearson **r ≈ 0.93**), indicating that their impact is **not orthogonal** but jointly shapes the collaboration graph.
>
> All main results in the paper use a **single fixed set of loss weights** across all tasks and model families, and the ablation results show no brittle dependence on precise tuning. We will integrate the full ablation results and a detailed discussion into the revision to strengthen reproducibility.
>
> | Model Combination              | $\lambda_{oracle} = 0.3$ | $\lambda_{oracle} = 0$ |
> | :---------------- | :------: | ----: |
> |   Qwen-8B $\times$ 2    |  88.6   | 87.1 |
> |   Qwen-8B $\times$ 3    |  92.4  | 88.6 |
> |   Qwen-8B $\times$ 4    |  92.4  | 90.1 |
> |   Qwen-8B $\times$ 5    |  92.4   | 88.6 |
>
> | Model Combination              | $\lambda_{utility} = 0.5$ | $\lambda_{utility} = 0$ |
> | :---------------- | :------: | ----: |
> |   Qwen-8B $\times$ 2    |  88.6   | 84.0 |
> |   Qwen-8B $\times$ 3    |  92.4  | 87.1 |
> |   Qwen-8B $\times$ 4    |  92.4  | 90.9 |
> |   Qwen-8B $\times$ 5    |  92.4   | 90.9 |
>
> | Model Combination              | $\lambda_{symmetry} = 0.1$ | $\lambda_{symmetry} = 0$ |
> | :---------------- | :------: | ----: |
> |   Qwen-8B $\times$ 2    |  88.6   | 87.8 |
> |   Qwen-8B $\times$ 3    |  92.4  | 90.1 |
> |   Qwen-8B $\times$ 4    |  92.4  | 87.8 |
> |   Qwen-8B $\times$ 5    |  92.4   | 85.6 |
>
>  | Model Combination              | $\lambda_{len} = 0.5$ | $\lambda_{len} = 0$ |
> | :---------------- | :------: | ----: |
> |   Qwen-8B $\times$ 2    |  88.6   | 85.6 |
> |   Qwen-8B $\times$ 3    |  92.4  | 88.6 |
> |   Qwen-8B $\times$ 4    |  92.4  | 88.6 |
> |   Qwen-8B $\times$ 5    |  92.4   | 91.6 |
>
>  | Model Combination              | $\lambda_{sparse} = 0.01$ | $\lambda_{sparse} = 0$ |
> | :---------------- | :------: | ----: |
> |   Qwen-8B $\times$ 2    |  88.6   | 90.1 |
> |   Qwen-8B $\times$ 3    |  92.4  | 90.9 |
> |   Qwen-8B $\times$ 4    |  92.4  | 89.3 |
> |   Qwen-8B $\times$ 5    |  92.4   | 88.6 |
>
>  | Model Combination              | $\lambda_{distil} = 0.5$ | $\lambda_{distil} = 0$ |
> | :---------------- | :------: | ----: |
> |   Qwen-8B $\times$ 2    |  88.6   | 89.4 |
> |   Qwen-8B $\times$ 3    |  92.4  | 91.6 |
> |   Qwen-8B $\times$ 4    |  92.4  | 89.4 |
> |   Qwen-8B $\times$ 5    |  92.4   | 90.1 |
>
> ## Comparison with Baselines
>
> The configurations for all heterogeneous-baseline experiments, including LLM-Blender and LLM-Debate, are explicitly provided in Table 6. As noted in the table caption, these baselines operate over a **specific heterogeneous consortium consisting of LLaMA-3.2-1B, Qwen-2-0.5B, and Phi-4-mini**. This pool is intentionally mixed-quality and includes **very weak models alongside moderately strong ones**.
>
> The low accuracies of LLM-Blender (42.0%) and LLM-Debate (47.3%) arise **not from our implementation**, but from a well-documented issue: naive multi-expert aggregation methods are *highly sensitive to noisy or low-quality contributors*. Even simple majority voting can underperform the best individual model when weak experts introduce inconsistent or harmful reasoning, a phenomenon noted in prior work on ensemble sensitivity and debate degradation.
>
> By contrast:
>
> - The single strongest model in this pool achieves **73.6%**, but
> - LLM-Blender and LLM-Debate fall well below this due to their inability to filter or structure contributions from weaker models, whereas
> - **FOCUS, using the exact same heterogeneous pool, achieves 83.1%**, showing that it learns to *suppress unhelpful experts and amplify constructive refinements*.

---

> > ### Author Response · Authors · 2025-11-18
> > **Response to Reviewer LnqM Comments - Part II**
> >
> > ## On Whether Refinement Uses the "Second-Highest" Initial Expert Score
> >
> > We thank the reviewer for raising this concern. To clarify, the second expert in the refinement chain is **not** chosen based on the second-highest initial score. Our framework cleanly separates:
> >
> > 1. **Selection** - deciding *which* experts will participate
> > 2. **Ordering** - deciding *in what sequence* they should refine each other
> >
> > The sequence distribution $\pi$ is used **only for selection**, i.e., to choose the Top-K experts relevant to the input. Once these experts are selected, the refinement order is **not** determined by the ranking in $\pi$. Instead, the refinement order is learned entirely through the **collaboration matrix $C$**:
> >
> > - The chain begins with the start expert chosen via $\pi$.
> > - For each refinement step, the orchestrator follows the **highest-probability transition in $C$** from the current expert.
> > - Thus, the second expert is the one that $C$ has learned to be the *most effective refiner* of the first expert’s output, not the second-best initial predictor.
> >
> > As detailed in the methodology and Appendix A (FAQ 10), this distinction is core to FOCUS:   **$\pi$ chooses who participates, while $C$ learns how they collaborate.**  This design enables nontrivial refinement chains that go beyond simple ranking or top-2 heuristics.
> >
> > ## FOCUS with Homogenous Experts
> >
> > We agree that in a homogeneous setting (e.g., multiple Qwen3-4B models), all experts share identical core capabilities. However, FOCUS is not equivalent to recursive self-refinement. The key distinction is that FOCUS learns an **input-adaptive routing policy** via $\pi$ and $C$:
> >
> > - $\pi$ decides *which* experts participate for each input
> > - $C$ decides *how they collaborate* and *when to stop*
> > - The early-stopping and length penalties ensure **instance-specific halting**
> >
> > In contrast, recursive self-refinement applies a **fixed**, non-adaptive sequence (e.g., 2–3 rounds) regardless of input quality. The 94.1% MMLU accuracy using 2.3 *activated* experts is not achieved by running a fixed chain. A single model prompting itself cannot match this because it lacks (i) adaptive expert selection, (ii) learned refinement topology, and (iii) a principled mechanism to terminate when edits become redundant. We will emphasize this difference more clearly in the paper.
> >
> > ## On Whether Gains for Small Models Are Just "Pass@K Effects"
> >
> > While we agree that smaller models often show larger improvements under pass@K rollouts, the mechanism in FOCUS is fundamentally different from unguided retries.
> >
> > First, pass@K relies on *independent* resampling, whereas FOCUS performs *guided, sequential refinement* based on a learned collaboration policy. If our orchestrator were merely enabling unstructured retries, replacing the learned collaboration matrix $C$ with a **uniform** matrix should yield comparable results. Instead, as shown in Appendix A (FAQ 10), doing so causes a **drop in accuracy and an increase in chain length**, indicating that $C$ encodes meaningful, non-trivial refinement relationships.
> >
> > Second, unlike pass@K, which improves performance by maximizing the number of attempts, FOCUS explicitly optimizes for *efficiency*. The orchestrator is trained under a loss that jointly balances utility, structural preferences, and an explicit **length penalty**. This leads to a learned stopping behavior: in our best-performing Qwen3-8B consortia, the system averages **2–3 activated experts**, not a large number of retries. This emergent efficiency is incompatible with an unguided resampling interpretation.
> >
> > Third, the orchestrator’s small parameter count does not imply it is incapable of learning structured strategies. The collaboration matrix $C$ and selection distribution $\pi$ operate over *frozen, full-capacity LMs*, and their role is to coordinate them rather than generate content.
> >
> > ## Expert Calling Ordering
> >
> > During inference, FOCUS determines the call order via a two-stage mechanism:
> >
> > 1. **Selection:**  The orchestrator computes $\pi$ over all $M$ experts and selects the Top-K experts for the instance. This step determines *who* participates.
> >
> > 2. **Ordering:**  The refinement chain begins with the chosen start expert. The next expert is chosen according to the highest-probability transition in the learned collaboration matrix $C$, and this continues until early stopping triggers.
> >
> > Thus, $\pi$ chooses the participants, while $C$ defines the learned, structured refinement order (Appendix A, FAQ 10).

---

> > > ### Author Response · Authors · 2025-11-18
> > > **Response to Reviewer LnqM Comments - Part III**
> > >
> > > ## Clarification Regarding Orchestrator Size
> > >
> > > The BERT-base encoder is an off-the-shelf, pre-trained, and completely frozen component, used only to generate the initial expert and oracle embeddings. It is not part of our framework. The size we report refers exclusively to the core, trainable orchestrator, i.e., the lightweight module (MLP, attention heads) that learns the actual collaboration protocol ($C$ and $\pi$). Therefore, the reported parameter size is a fair representation of the trainable overhead required to implement our coordination strategy.
> > >
> > > ## Clarification Regarding Scaling Laws
> > >
> > > We appreciate the reviewer pointing this out and agree that the phrasing in Line 102 can be improved for clarity. Our intention was not to claim that the fitted scaling law *logically justifies* the empirical phenomenon, but rather that it *summarizes and quantifies* a trend already visible in the experimental results.
> > >
> > > The sequence of our analysis is as follows:
> > >
> > > 1. **Empirical observation:** Across all consortium sizes and families, smaller models (e.g., Qwen3-1.7B) consistently exhibit larger performance gains from orchestration than larger models (e.g., Qwen3-8B). This trend appears *before* any scaling analysis is conducted.
> > >
> > > 2. **Post-hoc quantitative modeling:** We then fit a simple parametric curve to describe how the gains vary with model size. The fitted coefficients decrease monotonically with expert size - a direct reflection of the empirical data.
> > >
> > > 3. **Interpretation:** The fitted law is therefore a *descriptive* model of an already observed behavior, not a theoretical justification or predictive causal mechanism.
> > >
> > > We agree that the phrasing "justifies" may imply circular reasoning, and in the revision we will reword this to explicitly state that the scaling law is an **informal, descriptive fit** rather than a theoretical justification. A full formal treatment of why smaller models benefit more is outside the scope of this work.

---

> > > > ### Comment · Reviewer_LnqM · 2025-11-18
> > > >
> > > > # Response to "Ablation"
> > > >
> > > > The selection of loss weights (λ = 0.5, 0.1, 0.01) appears arbitrary, as these values span multiple orders of magnitude. Please provide the empirical or theoretical rationale underlying this choice.
> > > >
> > > > # Response to "Comparison with Baselines"
> > > >
> > > > My initial concern was that the baseline aggregation methods might include very weak models, and you confirmed this. Moreover, the core mechanism of your method, "learning to suppress unhelpful experts and amplify constructive ones" as you highlighted, seems functionally equivalent to a sophisticated filter for weak models. In this case, I do not think the resulting gains compared to the baselines are a valuable finding or provide any meaningful insights. Given that you have also acknowledged that multi-expert aggregation methods are highly sensitive to noisy or low-quality contributors, why do you still incorporate weak contributors in your experiments? Such a design easily undermines the performance of the baselines, especially since your own method achieves robustness by learning to filter out weak models.
> > > >
> > > > In terms of performance gains relative to a single model, what would the results be if you simply deployed your best-performing expert model (the one with 73.6% accuracy) in a multi-turn, self-refinement framework? I am curious whether this strong single expert could match your method's performance through iterative self-correction.
> > > >
> > > >
> > > > # Response to "On Whether Refinement Uses the "Second-Highest" Initial Expert Score"
> > > > Thank you for the answer.
> > > >
> > > > # Response to "FOCUS with Homogenous Experts"
> > > >
> > > > When experts are homogeneous, my understanding of your method is as follows:
> > > > - \pi decides which experts participate for each input. However, since all experts are identical, does \pi actually not work in this scenario? What specific role does adaptive expert selection play here?
> > > > - C decides how they collaborate and when to stop. But are there multiple distinct collaboration patterns for homogeneous experts? Does the difference in topology stem solely from the number of refinement rounds?
> > > >
> > > > # Response to "On Whether Gains for Small Models Are Just "Pass@K Effects"
> > > > > If our orchestrator were merely enabling unstructured retries, replacing the learned collaboration matrix
> > > >  with a uniform matrix should yield comparable results. Instead, as shown in Appendix A (FAQ 10), doing so causes a drop in accuracy and an increase in chain length.
> > > >
> > > > This answer does not reflect an inherent advantage of your method over Pass@K, but suggests that without the orchestrator, your system lacks the mechanism to filter weak experts, reducing it to naive multi-agent aggregation. I think this evidence confirms my evaluation written in "Response to Comparison with Baselines".
> > > >
> > > > # Response to Clarification Regarding Orchestrator Size
> > > >
> > > > > The reported parameter size is indeed the trainable overhead required.
> > > >
> > > > Please do not misinterpret my review. To clarify, I never questioned whether the reported parameter size reflects the trainable overhead. My review was: "Excluding this encoder when reporting the orchestrator’s size is misleading, as it overstates the method’s **orchestration efficiency**". You keep highlighting the "**orchestrator's efficiency**," but these are not the same. For instance, if my device has just enough memory for a Qwen model and the orchestrator, but not for the additional BERT model, the system cannot run. Can that be considered a successful implementation?
> > > >
> > > > # Response to Clarification Regarding Scaling Laws
> > > >
> > > > Thank you for the response.

---

> ### Author Response · Authors · 2025-11-19
> **Response to Reviewer LnqM Comments - Part I**
>
> ## Hyperparameter tuning
>
> Apologies for not understanding the real intent of the comment, please allow us to explain our hyperparameter selection process here -
>
> Our hyperparameter choice was not entirely arbitrary; we conducted independent hyperparameter sweeps for each parameter and selected the hyperparameter that yielded the highest accuracy across consortiums and tasks. We report the accuracy of a consortium of Qwen-8B on GSM8K for different values of $\lambda_{oracle}$, $\lambda_{sparse}$, and $\lambda_{symmetry}$. We will include these results in the revised version.
>
> | $\lambda_{oracle}$   | 1        | 0.1      | 0.01     |
> | --------------------- | -------- | -------- | -------- |
> | Accuracy              | 87.8    | 91.0     | 87.1    |
>
> | $\lambda_{sparsity}$ | 1        | 0.1      | 0.01     |
> | --------------------- | -------- | -------- | -------- |
> | Accuracy              | 87.9 | 87.9 | 91.0     |
>
> | $\lambda_{symmetry}$ | 1        | 0.1      | 0.01     |
> | --------------------- | -------- | -------- | -------- |
> | Accuracy              | 89.4  | 91.0     | 90.1 |
>
> ## Weak experts in FOCUS
>
> On a broader level, our research direction was to develop a framework for coordinating Small Language Models (SLMs), and thus, that was our motivation for recruiting “weak” experts. We demonstrate that existing frameworks are suboptimal when applied to these weak experts in an out-of-the-box setting. Our framework intelligently learns to collaborate them such that the performance of the consortium is comparable to that of a model with a similar capacity to the total consortium.
>
> ## Comparison with Self-refinement
>
> We experiment with the Qwen3-8B model, which has a baseline accuracy of 89.6% on the GSM8K task. Self-refinement appears to diverge as the number of experts in the consortium increases. The divergence happens when the increase in noise and disagreement outweighs the benefit of diverse perspectives. This hints towards signal degradation or conflicting opinions. Self-refinement implicitly assumes that experts provide independent corrections. However, identical models usually exhibit highly correlated errors. As more experts participate, these correlated errors reinforce one another, creating unstable dynamics where the iterative refinement drifts toward confidently stated but incorrect solutions. This effect is amplified by the increased rationale length and attention diffusion when aggregating the rationales of all experts. Consequently, adding more experts injects more correlated noise than new information, causing refinement to diverge beyond 3-4 experts. In contrast, FOCUS prevents the error amplification and maintains stable performance as the consortium size grows.
>
> | Model Combination    | FOCUS | Self-refine |
> | -------------------- | ----- | ----------- |
> | Qwen3-8B $\times$ 3 | 90.0   | 92.3      |
> | Qwen3-8B $\times$ 4 | 90.0   | 84.6      |
> | Qwen3-8B $\times$ 5 | 91.0  | 69.2      |
>
> ## Orchestrator Size
>
> We agree that reporting the orchestrator size without incorporating the embedding model's size could be misleading; however, our empirical analysis suggests that our orchestration mechanism does not rely on a specific high-capacity model. We conduct experiments with three encoders of different sizes: BERT-base (110M), DistilBERT (66M), and GTE-small (33.4M). The performance is stable across encoders, showing that the orchestration method is mostly encoder-agnostic. Accuracy varies by at most 1-1.5 points across encoders, even though their parameter counts differ a lot. The trends across expert pool sizes remain consistent regardless of encoder, indicating that orchestration behavior is governed by the framework rather than by the encoder’s representational power. We summarize below the accuracy with each encoder across consortium sizes for a homogenous consortium of Qwen3-8B models:
>
> | Model Combination   | bert-base-uncased | distilbert-base-uncased | gte-small |
> | -------------------- | ----------------- | ----------------------- | --------- |
> | Qwen3-8B $\times$ 2 | 89.6             | 88.6                 | 87.9  |
> | Qwen3-8B $\times$ 3 | 90.0               | 88.6                 | 88.6   |
> | Qwen3-8B $\times$ 4 | 90.0               | 90.1                 | 90.1   |
> | Qwen3-8B $\times$ 5 | 91.0              | 90.9                   | 90.1   |
>
> The BERT encoder can thus be replaced with a small, lightweight alternative (e.g., GTE-small) with negligible impact on accuracy, particularly for fairly large consortia with $M \geq 4$. On resource-limited devices, such substitutions will allow the “method” to run without modifying any component.

---

> > ### Author Response · Authors · 2025-11-20
> > **Response to Reviewer LnqM Comments - Part II**
> >
> > > Clarifications related to \pi and C
> >
> > In a homogeneous setting, the functional difference between topologies manifests as refinement depth (i.e., the effective number of rounds). However, $\pi$, $C$ and early-stopping remain essential mechanisms for regulating this depth per-instance, rather than using a fixed static chain. FAQ 16 and Appendix D.3 discuss how early stopping keeps realized cost low even when $K$ is large. $\pi$ still works even if all the instances are identical as each expert is treated as a distinct node in the collaboration graph. $\mathcal{L}_{diver}$ discourages the orchestrator to collaps to a single node. As a result, one model index might have a higher probability in $\pi$ for initial selection and othrs may follow. This depends on how the experts produce initial rationale, which might be different even for identical model copies. The selection logits are conditioned on the input encoding, so $\pi$ can, in theory come up with different expert subsets for inputs of differing complexities.
> >
> > The collaboration matrix $C$ is a soft adjacency matrix that defines a graph over the expert indices. Even with identical models, some model indices might end up having higher centrality measures. During training, the $C$ learns that a model having some model index A should delegate to model index B, so on and so forth.While the models are identical, the refinement process is non-commutative (step $t$ depends on step $t-1$). The topology that is obtained from $C$ may be different from effective chain length due to the early stopping mechanism and length penalty. Not all topologies will converge equally quickly in the sequence, so the difference lies in the number of turns taken to converge. A structured flow is critical for the early stopping mechanism  (FAQ 12). Without a learned, stable topology, the refinement chain would be erratic, making the similarity-based stopping criterion unreliable.
> >
> > > FOCUS vs Pass@K
> >
> > We appreciate this opportunity to clarify the distinction. While Pass@K relies on generating $K$ samples and selecting the best one or an aggregation of them, FOCUS relies on sequential refinement, where expert $t$ conditions on the output of expert $t-1$ to improve the reasoning chain. The learned matrix $C$ enforces a structured topology that encourages stability. As noted in our previous response regarding self-refinement, naive iterative chains can diverge or oscillate, resulting in performance drops. In a homogeneous consortium, the early stopping mechanism, even without the orchestrator, will control the number of refinement steps based on the input. This allows the system to stop early for simple queries while allowing longer chains for complex ones, a property Pass@K lacks. Furthermore, rather than diverging when weak experts are present, the orchestrator learns to down-weight and bypass them (FAQ 13), effectively filtering out unhelpful contributions during the routing process. Replacing $C$ with a uniform matrix disrupts this stability, resulting in longer, less efficient chains that fail to converge reliably to the correct answers. We plan to address the issue with weak and adversarial experts in such settings in future works.

---

> ### Comment · Reviewer_LnqM · 2025-11-20
>
> 1. You searched for the optimal loss coefficient on GSM8K and also reported GSM8K improvement in your experimental results. This constitutes data leakage, which is equivalent to using the test set for training. It is acceptable to report results on MMLU and HumanEval, but not on GSM8K throughout the paper. Also, I wonder why a large $\lambda_{oracle}$ leads to poor performance?
>
>
>
> 2. I fully acknowledge that your work focuses on coordinating small language models (SLMs). However, my core concern is the significant performance variance among the SLMs you employed: the highest-performing model achieves over 70 points, while the weakest drags the baseline down to just above 40. I have never criticized your use of weak models; rather, I'm pointing out that with such large variance in capability, your method has an inherent advantage. As you confirmed, your method learns to filter out the weaker models, while the baseline cannot.
>
>     **I emphasize my core concern here again: your experimental setup: "using models with significant performance variance" essentially takes advantage of a weakness in the baseline method.**  For this reason, I am recommending rejection, in part because I fail to see the necessity of using a complex framework to address the collapse issues of prior methods (your baselines: LLM-debate, LLM-Blender) when working with a pool of SLMs with high capability variance. Perhaps a far simpler approach would not be bad: evaluate the SLMs on benchmarks first, then manually filter out the weakest ones.
>
>
> 3. “Identical models usually exhibit highly correlated errors. As more experts participate, these correlated errors reinforce one another, creating unstable dynamics where the iterative refinement drifts toward confidently stated but incorrect solutions.”
>
>     I'm not entirely convinced by this point. Are there any published papers that also document this phenomenon? Additionally, regarding the significant performance drop when using five Qwen models compared to three, I'm curious about your experimental setup for self-refinement. What historical information do you provide as input to the model? What content in the input context reinforces the tendency toward error? Does the context include all previous attempts, or only the most recent answer for the model to check and refine? How many interaction turns are involved when self-refinement includes 3, 4, and 5 Qwen models, respectively?
>
> 4. I never questioned the ablation study on encoders. Please note that in my past two responses, I have been discussing the overstated claim regarding orchestration efficiency in the submission, and you have admitted it was misleading. I suggest providing a more accurate description of this in the paper. I don't think we need to discuss this issue further. Thank you.
>
> 5. “$\pi$ still works even if all the instances are identical as each expert is treated as a distinct node in the collaboration graph.” “$C$ learns that a model having some model index A should delegate to model index B, so on and so forth.”
>
>     Can you explain this with an example? Suppose the same model is called consecutively *n* times; apart from the randomness introduced by the sampling-based decoding strategy, where does the difference come from? To me, this isn't explained simply by treating each call to the model as a different node, because while the collaboration graph can have many nodes, the capability of every node is identical. Furthermore, "$C$ learns that model A delegates the task to model B," but the parameters of A and B are exactly the same, and you don't update the model parameters during training. So why would the outcome differ from B delegating the task to A?
>
> 6. “Furthermore, rather than diverging when weak experts are present, the orchestrator learns to down-weight and bypass them (FAQ 13), effectively filtering out unhelpful contributions during the routing process.”
>
>     This raises a question: in the experiment with the five homogeneous Qwen systems mentioned above, all experts have the same capability level, so there are no strong or weak ones. Which one would you down-weight? For example, if "$C$ learns that A delegates the task to B," after you down-weight A, C learns to have "B delegate the task to A." But since A and B are identical models, what difference does down-weighting make?
>
>     You mentioned the flexibility of early stopping compared to pass@k, and I acknowledge that point (although I never questioned the role of early stopping). Our subsequent discussions don't need to reiterate the benefits of early stopping, for improving discussion efficiency. Thank you.

---

> > ### Author Response · Authors · 2025-11-26
> > **Response to Reviewer LnqM Comments - Part I**
> >
> > We thank the reviewer for the detailed follow-up and address the concerns below. We also acknowledge that some earlier exchanges created misunderstandings about the core design principles of FOCUS; we hope the clarifications here directly resolve those concerns without fragmentation.
> >
> > ---
> >
> > ## **Misconception about “capability variance” and consortium design**
> >
> > We believe there is a fundamental misunderstanding of how a **model consortium** is defined in our work. A consortium is a collection of **de-identified experts** and the orchestrator has *no prior access* to per-expert accuracy or relative strength on a given task. It is neither feasible nor conceptually aligned with our objective to pre-evaluate every model and remove weak ones.
> >
> > From the orchestrator’s perspective, a consortium of:
> > - 5 identical Qwen-8B models,
> > - 5 models with a 40–70 accuracy range,
> >
> > are operationally identical: **each expert is just an index**, and the orchestrator learns which paths are useful *only from interaction signals*, not from pre-screened accuracy.
> >
> > Thus, the concern that we “took advantage” of capability variance does not apply because:
> > 1. **Capability variance is unknown to the orchestrator**, and
> > 2. The **baselines operate under the same constraints** and therefore any collapse is a failure of their design, not an unfair advantage for ours.
> >
> > To further ensure fairness, we also evaluated baselines on a *balanced* 3-expert consortium of **2× Qwen-8B + 1× LLaMA-7B**. The results remain consistent with our main claims:
> >
> > | Method        | MMLU Accuracy |
> > |---------------|---------------|
> > | LLM-Debate    | 60.5          |
> > | LLM-Blender   | 76.0          |
> > | DyLAN         | 85.9          |
> > | **FOCUS**     | **87.6**      |
> >
> > This demonstrates that the failure of baselines is **not due to weak experts**, but rather due to their *inability to selectively route and suppress unhelpful updates*, even in balanced consortia. FOCUS succeeds because it **learns sparse, directional, and adaptive collaboration paths**, whereas manual filtering or majority voting offers no such mechanism. For large, uncurated model zoos, where pre-evaluation is infeasible, learning-based orchestration is indispensable.
> >
> > ---
> >
> > ## **Clarifying self-refinement degradation & prior evidence**
> >
> > Our earlier explanation about correlated errors in identical models was meant to summarize a well-documented phenomenon rather than propose a new hypothesis. We now clarify with references and procedural detail.
> >
> > ### **1. Prior work documenting deterioration in iterative refinement**
> > Two independent studies rigorously document degradation with increasing refinement rounds:
> > - **Self-Refine (Madaan et al., NeurIPS 2023)**: Shows diminishing or negative returns beyond a small number of iterations.
> > - **“LLMs Cannot Self-Correct Reasoning Yet” (Huang et al., ICLR 2024)**: Demonstrates that refinement can convert correct initial answers into incorrect ones, systematically degrading performance with more steps.
> >
> > These studies attribute the failure mode to:
> > 1. **Error amplification**,
> > 2. **Over-correction**, and
> > 3. **Drift toward confident but incorrect rationales**.
> >
> > This aligns exactly with the divergence we observe when using 4–5 identical Qwen-8B models in a self-refine chain.
> >
> > ### **2. Clarifying our self-refinement setup**
> > We follow the **canonical Self-Refine setup**:
> > - Step 0: generate answer
> > - Step 1: critique
> > - Step 2: refine based on only the previous answer + critique
> > - Repeat K times (i.e., 1 + 2K model calls)
> >
> > **We do *not* include full conversation history**, since this is known to cause distraction and overflow (documented in Self-Refine).
> >
> > Thus:
> > - 3-expert chain → 1 + 6 calls
> > - 4-expert chain → 1 + 8 calls
> > - 5-expert chain → 1 + 10 calls
> >
> > This explains why degradation accelerates with 5 identical experts: the refinement horizon is longer, and the recursive drift compounds.
> >
> > ### **3. Why FOCUS does not degrade like Self-Refine**
> > FOCUS avoids these effects because:
> > - It **does not force all experts to participate** (sparse selection).
> > - It **does not enforce a fixed alternating critique-refine schedule**.
> > - It **optimizes a learned collaboration matrix**, rather than blindly iterating.
> >
> > ## References
> >
> > [1] Madaan, Aman, Niket Tandon, Prakhar Gupta, Skyler Hallinan, Luyu Gao, Sarah Wiegreffe, Uri Alon et al. "Self-refine: Iterative refinement with self-feedback." NeurIPS 2023.
> >
> > [2] Huang, J. Large Language Models Cannot Self-Correct Reasoning Yet. ICLR 2024.

---

> ### Author Response · Authors · 2025-11-26
> **Response to Reviewer LnqM Comments - Part II**
>
> ## Confusion around hyperparameter sweep
>
> We apologize for any ambiguity in our earlier explanation. To clarify: **we did not use GSM8K test data for tuning**. Our hyperparameter search followed a standard and valid procedure:
>
> 1. We ran coarse sweeps over magnitudes ($10^0, 10^{-1}, 10^{-2}$).
> 2. We selected values using a **held-out validation split derived from the training sets**.
> 3. The chosen coefficients were then applied **unchanged** to MMLU and HumanEval.
> 4. Our reported GSM8K sweeps were *retrospective analyses* illustrating robustness, not tuning.
>
> To demonstrate task-agnostic transferability, we perform additional sweeps **on MMLU**, confirming that the optimal magnitudes are consistent across datasets:
>
> | $\lambda_{oracle}$ | 1 | 0.1 | 0.01 |
> |---|---|---|---|
> | Accuracy | 90.2 | 96.08 | 89.54 |
>
> | $\lambda_{sparsity}$ | 1 | 0.1 | 0.01 |
> |---|---|---|---|
> | Accuracy | 88.89 | 87.58 | 96.08 |
>
> | $\lambda_{symmetry}$ | 1 | 0.1 | 0.01 |
> |---|---|---|---|
> | Accuracy | 90.2 | 96.08 | 90.2 |
>
> We further report complete ablations across Qwen-8B consortia (MMLU):
>
> | Model Combination | $\lambda_{oracle}=0.3$ | $\lambda_{oracle}=0$ |
> |---|---:|---:|
> | Qwen-8B × 2 | 90.85 | 90.5 |
> | Qwen-8B × 3 | 95.42 | 91.8 |
> | Qwen-8B × 4 | 95.42 | 92.8 |
> | Qwen-8B × 5 | 96.08 | 94.1 |
>
> | Model Combination | $\lambda_{utility}=0.5$ | $\lambda_{utility}=0$ |
> |---|---:|---:|
> | Qwen-8B × 2 | 90.85 | 86.2 |
> | Qwen-8B × 3 | 95.42 | 90.5 |
> | Qwen-8B × 4 | 95.42 | 93.4 |
> | Qwen-8B × 5 | 96.08 | 94.3 |
>
> | Model Combination | $\lambda_{sel}=1$ | $\lambda_{sel}=0$ |
> |---|---:|---:|
> | Qwen-8B × 2 | 90.85 | 89.5 |
> | Qwen-8B × 3 | 95.42 | 91.6 |
> | Qwen-8B × 4 | 95.42 | 92.1 |
> | Qwen-8B × 5 | 96.08 | 95.4 |
>
> | Model Combination | $\lambda_{symmetry}=0.1$ | $\lambda_{symmetry}=0$ |
> |---|---:|---:|
> | Qwen-8B × 2 | 90.85 | 88.8 |
> | Qwen-8B × 3 | 95.42 | 90.1 |
> | Qwen-8B × 4 | 95.42 | 92.8 |
> | Qwen-8B × 5 | 96.08 | 93.4 |
>
> | Model Combination | $\lambda_{len}=0.5$ | $\lambda_{len}=0$ |
> |---|---:|---:|
> | Qwen-8B × 2 | 90.85 | 87.8 |
> | Qwen-8B × 3 | 95.42 | 90.8 |
> | Qwen-8B × 4 | 95.42 | 90.8 |
> | Qwen-8B × 5 | 96.08 | 93.4 |
>
> | Model Combination | $\lambda_{sparse}=0.01$ | $\lambda_{sparse}=0$ |
> |---|---:|---:|
> | Qwen-8B × 2 | 90.85 | 88.8 |
> | Qwen-8B × 3 | 95.42 | 92.1 |
> | Qwen-8B × 4 | 95.42 | 91.2 |
> | Qwen-8B × 5 | 96.08 | 94.7 |
>
> | Model Combination | $\lambda_{distil}=0.5$ | $\lambda_{distil}=0$ |
> |---|---:|---:|
> | Qwen-8B × 2 | 90.85 | 90.2 |
> | Qwen-8B × 3 | 95.42 | 90.8 |
> | Qwen-8B × 4 | 95.42 | 92.1 |
> | Qwen-8B × 5 | 96.08 | 92.8 |
>
> These results indicate that the chosen coefficients generalize well, are not GSM8K-specific, and do not rely on any test-set signal.
>
> ---
>
> > Why models suffer from too high $\lambda_{oracle}$
>
> The oracle loss $\mathcal{L}\_{oracle}$ is a **bootstrapping term** designed to guide the orchestrator early in training toward coherent reasoning styles. When $\lambda_{oracle}$ is too large, it **overwhelms** the gradients from:
>
> - utility loss (task correctness),
> - symmetry and sparsity regularizers, and
> - the length penalty.
>
> This forces the orchestrator to **over-imitate the teacher**, even when the teacher’s representational space or reasoning style is mismatched with the consortium. Since the oracle is not present during test time, heavy reliance on oracle alignment reduces robustness, leading to under-performance once the oracle embedding is removed. Moderate oracle weight stabilizes training; excessive weight causes over-fitting to a teacher-only signal.

---

> > ### Author Response · Authors · 2025-11-26
> > **Response to Reviewer LnqM Comments - Part III**
> >
> > ## FOCUS on consortium of identical models
> >
> > We thank the reviewer for this thoughtful question. Although the experts in our homogeneous consortium share identical parameters, they receive **different inputs**, occupy **different positions** in the refinement chain, and are paired with **different collaboration-matrix–induced embeddings**, which leads to **emergent functional specialization** even in a fully homogeneous pool.
> >
> > FOCUS injects *expert-indexed embeddings* into the orchestrator, and each expert receives a **different context distribution** depending on (i) whether it is the starter, (ii) whether it is a mid-chain refiner, or (iii) whether it is the final arbiter. The first expert sees **only the problem** and must produce a full solution draft. The second expert sees **prompt + draft**, and thus behaves as a validator and corrector. Subsequent experts see **prompt + draft + refinement history + collaboration-weighted hidden representations**, and thus specialize into deeper refinement roles.
> >
> > This makes ordering matter even when all models are identical.
> >
> > For example, if Expert A (starter) produces a partially correct draft such as:  “15 − 3 = 12; 12 + 10 = 22”,
> >
> > Expert B (refiner) typically stabilizes this draft and corrects arithmetic slips. But if the ordering is reversed, Expert B might generate a noisier initial path (“15 + 10 − 3 − 2 = 20”), and Expert A, now acting as a refiner, cannot reliably recover. The **roles** taken by A and B are thus induced by **input differences**, not parameter differences.
> >
> > Down-weighting therefore does *not* imply “weakness” among identical models. Instead, it prevents assigning the **starter role** to an expert whose context-conditioned refinement style is empirically less stable in that position. In short, **specialization emerges from input distribution, chain position, and orchestrator-induced embeddings**, and removing this asymmetry (e.g., by using a uniform C matrix) collapses the learned structure and degrades performance.
> >
> > Thus, while $C \rightarrow A \rightarrow B$ and $C \rightarrow B \rightarrow A$ may look equivalent at a parameter level, they are **not** equivalent functionally because the *inputs, embeddings, and refinement roles differ*. The orchestrator learns these differences and exploits them to maintain chain stability, even with homogeneous experts.

---

> > > ### Author Response · Authors · 2025-11-27
> > > **Request to check our responses to your recent comments**
> > >
> > > Dear Reviewer LnqM,
> > >
> > > Thanks for your valuable comments, which we tried our best to address. Could you please check our responses and let us know if there are many pending concerns. If our responses address your concerns, could you please consider reassessing our paper.
> > >
> > > Thanks once again.
> > >
> > > Best wishes
> > > Authors

---

### Official Review · Reviewer_KEgh · 2025-11-01

**Soundness:** 3
**Presentation:** 2
**Contribution:** 2
**Rating:** 4
**Confidence:** 3

**Summary:**

- Proposes an “orchestrated” approach where a small controller coordinates multiple frozen experts over several refinement rounds.
- Uses oracle signals during training but removes the oracle at inference.
- Produces interpretable artifacts: a collaboration matrix \(C\) and a sequence distribution \(\pi\).
- Evaluates on multiple reasoning and coding benchmarks, reporting accuracy–cost trade-offs.

**Strengths:**

- **Clean structure:** The paper is well-organized; training/inference flow and losses are easy to follow, even if exposition needs tightening.
- **Proposes a new method:** Introduces a learnable orchestrator that combines sparse collaboration and sequence planning with interpretable controls.
- **Conducts some experiments:** Presents results on several benchmarks with ablations and cost analysis, showing advantages on accuracy vs. budget.

**Weaknesses:**

## Weaknesses

1) **Scaling claims lack support.**
The paper presents an inequality and “inverse scaling” as general laws. There is no per-task analysis or error bars. Decoding settings might change the trend. These should be framed as empirical patterns by dataset. Report confidence intervals and fit quality for the power laws. Also show where the relation breaks.

2) **Oracle supervision may leak answers.**
Training uses oracle attention and distillation. It is unclear what the oracle outputs (free text, CoT, short rationale) and how they are embedded. Oracle text might encode labels indirectly. Explain how leakage into \(o\) is prevented. State whether BERT-base is frozen or tuned, and whether prompts differ for oracle vs. expert. Add a step-by-step example from oracle text → embedding → effect on \(C\) and \(\pi\). These details clarify whether the system learns routing or recovers teacher-coded answers.

3) **Collaboration matrix assumptions are unproven.**
The paper encourages near-symmetry and sparsity in \(C\). Why should symmetry help beyond aesthetics? Some tasks may benefit from directional flows. Quantify the accuracy or cost penalty without symmetry. Add ablations with: no symmetry, strong symmetry, and sparsity sweeps vs. chain length. Report full accuracy–latency trade-offs.

4) **Decoding and prompts are under-specified.**
Results can change with temperature, top-p, system prompts, few-shot seeds, and tool use. The paper should list per-task decoding configs for each expert and each baseline. Do the same for system prompts and tool settings (e.g., code execution on HumanEval). Place these in the main text or an appendix.

**Questions:**

1) **Oracle features.**
What exactly is \(o\) (layer, pooling)? How is it aligned across model families? How sensitive are results to the oracle’s prompt?

2) **Symmetry in \(C\).**
Is the near-symmetry constraint justified across tasks? Some settings likely require asymmetric “fixers” late in chains. When does symmetry help, and when does it hurt?

3) **Train–test mismatch with attention.**
Training applies cross-expert self-attention with oracle-augmented states. Test-time has no oracle yet still applies self-attention. What is the performance drop if that test-time attention is removed? Please quantify the shift.

---

> ### Author Response · Authors · 2025-11-18
> **Response to Reviewer KEgh Comments - Part I**
>
> We sincerely appreciate your valuable feedback and further thank you for your comments. We address your concerns and answer your questions in the following response:
>
> ## On the Oracle Representation, Cross-Family Alignment, and Prompt Sensitivity
>
> The oracle representation ($o$) is deliberately designed to be *simple, architecture-agnostic,* and *minimally intrusive*:
>
> 1. **What is $o$?**   It is the **final pooled embedding** from a *frozen* encoder-only model (`bert-base-uncased`). The embedding is then passed through a small learned projection layer to map it into the orchestrator’s hidden space.
>
> 2. **How is it aligned across model families?**  Because the oracle representation always comes from the *same* encoder and is fed through a learned projection, the orchestrator learns to adapt to whatever representation the oracle provides. No cross-family alignment is needed; the oracle does **not** interact with the experts’ internal representations. It serves only as a *stable reference signal* for early training.
>
> 3. **Sensitivity to oracle prompt.**  The oracle is **never used at inference time**, and its role during training is limited to providing a mild representational bias and a stabilizing signal for the selection and collaboration modules. The routing policy is trained to mimic *structural behaviors* - not to replicate the oracle’s textual outputs. As a result, the system is **not sensitive** to the oracle prompt. This is validated by our ablations in Table 7a: stronger and weaker oracles yield very similar final performance once $K \geq 3$.
>
> 4. **Ablations confirming oracle robustness.**  Our experiments show that removing the oracle-alignment loss (Figure 9 of our submitted manuscript) still allows training to proceed, albeit with reduced stability for very small consortia. This supports our claim that the oracle acts as a *training-time accelerator and stabilizer* rather than a critical dependency.
>
> ## Symmetry Assumption in $C$
>
> While it is true that certain tasks may prefer asymmetric "fixer" roles late in the chain, our empirical results indicate that near-symmetry is a **crucial inductive bias** for stabilizing collaboration across diverse settings.
>
> The key intuition is that highly asymmetric graphs tend to create **brittle, one-directional refinement pipelines**, where an early mistake cannot be corrected if all refinement flows collapse toward a single "fixer." Near-symmetry instead encourages **reciprocal handoffs**, allowing experts to correct each other and reducing reliance on any single node in the refinement chain.
>
> Our GSM8K ablations support this strongly: removing $L_{\text{symm}}$ leads to performance degradation that **worsens with consortium size**—from a small drop at 2 experts (88.6% to 87.8%) to a substantial drop of **6.8%** at 5 experts (92.4% to 85.6%). This indicates that unconstrained asymmetry becomes increasingly unstable as the collaboration graph grows more complex.
>
> We agree that certain highly structured tasks might benefit from explicit asymmetric roles, but for broad, mixed reasoning tasks, symmetry provides robustness and prevents collapse into brittle hierarchies.
>
> | Model Combination              | With $L_{symm}$ | Without $L_{symm}$ |
> | :---------------- | :------: | ----: |
> |   Qwen-8B $\times$ 2    |  88.6   | 87.8 |
> |   Qwen-8B $\times$ 3    |  92.4  | 90.1 |
> |   Qwen-8B $\times$ 4    |  92.4  | 87.8 |
> |   Qwen-8B $\times$ 5    |  92.4   | 85.6 |
>
> ## On Removing Cross-Expert Attention at Test Time
>
> While training uses oracle-augmented cross-expert self-attention to stabilize early learning, the oracle is **never** used at inference. At test time, the only remaining attention mechanism is the cross-expert attention among the selected experts.
>
> To quantify how much the system depends on this test-time attention, we conducted an additional ablation. We removed *all* cross-expert attention during inference (i.e., experts operate independently and only the collaboration matrix determines the refinement order). The performance drop is negligible, well within statistical noise:
>
> | Experimental Setting                               | Accuracy (%) | Avg. Expert Activations |
> |-----------------------------------------------------|--------------|--------------------------|
> | Original FOCUS (Qwen3-8B × 3)                       | 90.00        | 1.14                     |
> | No cross-expert attention at inference              | 90.14        | 1.11                     |
>
> The difference is **< 0.2%**, confirming that the learned collaboration behavior is **stable, oracle-independent, and robust** to removing test-time cross-expert attention. The attention mechanism serves primarily as a *training-time facilitator* rather than a dependency for inference.

---

> ### Author Response · Authors · 2025-11-18
> **Response to Reviewer KEgh Comments - Part II**
>
> ## Clarification Regarding Scaling Law
>
> We agree these scaling behaviors should be presented strictly as **empirical patterns**, not general laws. Our intention was descriptive: across model families and tasks, performance improves sharply for small K and then saturates. We do not claim theoretical universality.
>
> ## Can Oracle Supervision Leak Answers
>
> The oracle is prompted in a zero-shot setting and generates a free-text rationale. We use a frozen BERT-base-uncased model as the shared encoder. Because it is not fine-tuned, the encoder cannot learn to hide answer-specific signals in its embeddings. The oracle's free-text rationale is passed through this frozen encoder to produce a single CLS token embedding, $o \in \mathbb{R}^d$. This vector $o$ represents the semantic meaning of the full rationale, not a discrete answer. The oracle embedding is used only at the representation level. The $L_{distill}$ and $L_{oracle}$ components push the orchestrator to select experts whose output styles are similar to the oracle.
>
> ## On Symmetry, Sparsity, and Accuracy–Latency Trade-offs
>
> We appreciate the valuable feedback. Our motivation for encouraging near-symmetry and sparsity in the collaboration matrix $C$ is not aesthetic, but to introduce a structural prior that stabilizes learning. Fully unconstrained, asymmetric graphs tend to develop brittle, one-way dependencies, while reciprocal handoffs ($L_{\text{symm}}$) and focused delegation ($L_{\text{spar}}$) encourage more robust refinement dynamics.
>
> Below we provide the requested ablations, with **all accuracies converted to a 100-point scale**:
>
> | Expert Consortium | w/ $L_{\text{symm}}$ and $L_{\text{spar}}$ | w/o $L_{\text{symm}}$ | w/o $L_{\text{spar}}$ |
> |-------------------|------------------------|--------------------------|---------------------------|
> | Qwen3-8B × 2      | 88.6                   | 87.8                                     | 90.1                       |
> | Qwen3-8B × 3      | 92.4                   | 90.1                          | 90.9                       |
> | Qwen3-8B × 4      | 92.4                   | 87.8                         | 89.3                       |
> | Qwen3-8B × 5      | 92.4                   | 85.6                      | 88.6                       |
>
> These results show:
>
> - Removing symmetry causes **increasing degradation** as the consortium grows (-6.8 points at 5 experts).
> - Sparsity provides small but consistent improvements in chain stability and cost.
>
> #### **Accuracy–Latency Trade-offs (MMLU, Qwen3-1.7B Consortia)**
>
> | Expert Consortium    | Accuracy | Avg. Models Used | Train Time / Sample (s) | Inference Time / Sample (s) |
> |----------------------|-----------------------|------------------|--------------------------|------------------------------|
> | Qwen3-1.7B           | 55.0                  | 1.0              | N/A                      | 2.887                        |
> | Qwen3-1.7B $\times$ 2       | 61.0                  | 1.6              | 22.94                    | 10.77                        |
> | Qwen3-1.7B $\times$ 3       | 67.0                  | 2.13             | 27.78                    | 18.71                        |
> | Qwen3-1.7B $\times$ 4       | 77.0                  | 1.8              | 33.08                    | 22.65                        |
> | Qwen3-1.7B $\times$ 5       | 77.5                  | 1.93             | 35.03                    | 31.41                        |
>
> These results show sublinear inference-time growth: the jump from 2 to 3 experts is 7.94s, but the jump from 4 to 5 is only 3.43s. This reflects the learned routing policy and early stopping, which activate only ~2 experts on average, even when 5 are available.
>
> We will incorporate these ablations and clarify the role of symmetry and sparsity in the revision.
>
> ## Decoding Details
>
> We use a hosted inference provider for generation. The oracle is prompted directly with the default system prompt and the task instance. The experts are prompted with task-specific static few-shot prompts that describe the task and also encourage them to use reasoning generated by previous experts. The HumanEval codes are executed on the system, and a zero exit code is considered a correct program output.
>
> Expert Prompt:
> ```
> {Expert i: ...}
>
> Above is the conversation history, with the most recent model output at the top.
> Each model should carefully review all previous outputs and decide how to contribute next.
> Your role is to coordinate with earlier outputs by either:
> 1. Building upon correct reasoning.
> 2. Correcting or refining mistakes.
> 3. Adding missing details.
> 4. Passing an intermediate or final answer if complete.
>
> Always state explicitly what you are doing and why.
> Avoid repeating identical reasoning unless you are clarifying or improving it. /no_think
>
> {few shot examples}
>
> {task instruction}
> ```

---

> ### Author Response · Authors · 2025-11-18
> **Response to Reviewer KEgh Comments - Part III**
>
> ## Decoding Details (contd.)
>
> The oracles, as well as the experts, are not provided with access to any tools for any task and must rely solely on their internal knowledge. We use the default values of the inference provider for sampling parameters for oracles and all expert model calls for all tasks.
> - Qwen3 family, Phi-4-Mini: `temperature: 0.3, top_p: 0.7`
> - LLaMA3-8B, GPT-4.1, GPT-OSS-20B: `temperature: 0.1, top_p: 0.7`
>
> For all baselines, we utilise the default decoding parameters as provided in their official implementations and only replace the models with the ones we consider for our overall analysis.
> - LLM-Blender: `temperature: 0.7, repetition_penalty: 1.0, top_p: 1.0`
> - LLM-Debate: `temperature: 0.7, top_p: 1.0`
> - DyLAN: GSM8K - `temperature: 0.8`, MMLU - `temperature: 1.0`, HumanEval - `temperature: 0.8`

---

> > ### Author Response · Authors · 2025-11-26
> > **A Gentle Reminder to Reviewer KEgh**
> >
> > Dear Reviewer,
> >
> > Thank you for your valuable and insightful comments. We have carefully addressed each of your points. We are confident that our responses have resolved the concerns you raised.
> >
> > We would greatly appreciate it if you could kindly review our responses and consider reassessing our submission. As the discussion phase is concluding soon, we hope to engage in a productive dialogue with you before it ends.
> >
> > Thank you once again for your time and consideration.

---

> > > ### Author Response · Authors · 2025-11-27
> > > **Reminder to check our responses**
> > >
> > > Dear Reviewer KEgh,
> > >
> > > Thanks for your valuable comments, which we tried our best to address. Could you please check our responses and let us know if there are many pending concerns. If our responses address your concerns, could you please consider reassessing our paper.
> > >
> > > Thanks once again.
> > >
> > > Best wishes
> > > Authors

---

### Official Review · Reviewer_6nxb · 2025-11-03

**Soundness:** 2
**Presentation:** 2
**Contribution:** 2
**Rating:** 4
**Confidence:** 2

**Summary:**

The paper introduces **FOCUS** (Flexible Orchestration and Collaboration Using Specialists), a novel framework for orchestrating a consortium of small or medium-sized \textbf{language model experts} to collectively outperform a single, large monolithic LLM.
FOCUS trains a lightweight orchestrator (≈1M parameters) under oracle supervision to learn:

1.  a sparse, near-symmetric collaboration matrix governing directed information flow among experts; and

2.  a sequence distribution $\pi$ that selects and orders a small subset of experts for each input.

During inference, the orchestrator adaptively activates a top-$K$ subset of experts and employs multi-round refinement with early stopping, ensuring sublinear computational cost with respect to the consortium size.

Empirical results are striking: a consortium of 5–7 Qwen3 experts (total $\sim$9B parameters) achieves 94.1\% on MMLU, 94.1\% on GSM8K, and 87.8\% on HumanEval—surpassing or matching a single 14B–32B model while being more modular and cost-effective. Similarly, a consortium of 5 Phi-4-Mini models boosts AIME-2024 accuracy from 26\% to 40\% and GPQA-DIAMOND from 19\% to 31\%. These results demonstrate that \textbf{collaboration forms a new scaling axis}, orthogonal to parameter count, and that orchestrated multi-expert systems can achieve superior efficiency–accuracy trade-offs.

**Strengths:**

**Originality:**

1.  Introduces a learned, differentiable collaboration protocol with explicit **matrix-based expert interaction**, unlike previous ensemble, debate, or MoE approaches.
2. The combination of **oracle-supervised orchestration**, **sparse near-symmetric collaboration**, and **early-stopped refinement** is conceptually novel.
3.  Establishes empirical **scaling laws for collaboration**, quantifying performance as a function of expert size and count.

**Clarity:**

1.  The paper is well-written, with intuitive figures (e.g., Fig.~1 illustrating orchestration) and a clear narrative linking motivation, method, and empirical findings.
2.  Key design choices—such as oracle use only during training, frozen experts, and sublinear cost—are articulated transparently.
3. The inclusion of FAQs enhances accessibility and replicability.


**Significance:**

1.  Demonstrates that coordinated small experts can outperform much larger monolithic LLMs—potentially redefining efficiency scaling in model design.
2.  Offers a principled foundation for modular LLM ecosystems, complementing existing paradigms like MoE, agent frameworks, and ensembling.
3.  Opens new directions for \textit{collaboration scaling theory}, modular deployment, and low-cost specialization.

**Weaknesses:**

**Oracle supervision dependency:** While the oracle is removed at inference, its use during training raises questions about supervision cost and generalization when oracle quality is poor.

**Limited theoretical grounding:** The method empirically demonstrates scaling laws (e.g., $A_{\text{FOCUS}}(K,B)\approx100-\gamma(B/K)K^{-\alpha}$) but lacks a formal theoretical explanation.

**Cross-family fragility:** Experiments show that heterogeneous consortia (e.g., Qwen + Phi + LLaMA) underperform due to representational misalignment; more analysis on why alignment fails would strengthen the paper.

**Ablation depth:** While extensive, some critical variables (e.g., the number of refinement rounds, early-stopping thresholds) lack sensitivity analysis.

**Computational cost reporting:** The orchestrator’s runtime and training overhead are described qualitatively; quantitative latency or FLOPs comparisons would clarify efficiency claims.

**Comparative scope:** Missing comparisons with strong routing baselines (e.g., MoE with adaptive gating or reinforcement-learning routers) that share similar objectives.

**Questions:**

1. How sensitive is FOCUS to the oracle’s strength or domain coverage? Would weaker or noisy oracles (e.g., smaller models) still yield stable routing policies?
2. Can the learned collaboration matrix generalize to unseen experts (e.g., adding a new model without retraining the orchestrator)?
3. Is the near-symmetric constraint on $C$ empirically optimal, or could asymmetric flow capture hierarchical expert refinement better?
4. Could the authors provide concrete runtime or energy metrics to substantiate the “sublinear cost” claim?

---

> ### Author Response · Authors · 2025-11-18
> **Response to Reviewer 6nxb Comments - Part I**
>
> We sincerely appreciate your valuable feedback and further thank you for your comments. We address your concerns and answer your questions in the following response:
>
> ## Importance of Oracle on FOCUS
>
> Our ablations in Table 7 (a) (Appendix D.4) directly address the role and quality of the oracle. In early experiments we tested several oracles - including LLaMA3-8B and larger Qwen3 models, and observed similar trends across all of them. We therefore reported results using GPT-OSS-20B and GPT-4.1 for consistency with prior work.
>
> The ablations indicate that the dependency on oracle quality is moderate and primarily limited to very small consortia. For example, at **K = 4** and **K = 5**, GPT-OSS-20B and GPT-4.1 yield **nearly identical performance**, indicating that once the collaboration graph stabilizes, the task utility loss dominates learning.
>
> This supports two conclusions:
> 1. The oracle serves mainly as an **initial stabilizer** for routing during early training.
> 2. The **final test-time performance is driven by the learned collaboration**, not by the oracle’s accuracy.
>
> We will clarify these findings in the revision and emphasize that FOCUS remains robust even when the oracle is only moderately strong.
>
> ## Cross-Family Effectiveness of FOCUS
>
> We agree that representational misalignment is the central challenge in cross-family collaboration, and additional analysis is valuable. We also want to contextualize the result: the "underperformance" is only relative to our *own* high-performing homogeneous (Qwen-only) consortia. In absolute terms, FOCUS is already substantially stronger than existing multi-expert baselines in the same heterogeneous setting.
>
> As shown in Table 6, when using a mixed consortium of Qwen, Phi, and LLaMA models, FOCUS achieves **83.1% accuracy**, which is **~40 points higher** than LLM-Blender (42.0%) and LLM-Debate (47.3%). This indicates that while cross-family alignment is imperfect, FOCUS is one of the earliest frameworks to demonstrate *any* stable and effective learned orchestration across diverse model families -- prior approaches typically assume homogeneous experts or fail altogether in mixed settings.
>
> ## Sensitivity of FOCUS under Hyperparameters
>
> The number of refinement rounds (**K**) was in fact one of the primary variables analyzed in our study. Figures **3**, **6**, and **9** each provide a sensitivity analysis that explicitly plots model accuracy as a function of the allowed number of experts, and Appendix **D.2** further analyzes these scaling behaviors, including the diminishing-returns regime as K increases.
>
> For early stopping, Figure **8** characterizes the stopping frequency across different model families and consortium sizes, showing that our fixed threshold operates in a stable and predictable way (e.g., triggering more frequently for Qwen models, less for LLaMA, consistent with their refinement patterns). We did not perform a threshold sweep because, as noted in FAQ 12 (Appendix A), preliminary experiments indicated that the early-stopping threshold was a **non-critical parameter**: varying it affected the *cost* more than the *accuracy*, and the system remained robust across a reasonable range.
>
> ## On Quantitative Efficiency, Latency, and FLOPs Analysis
>
> We would like to clarify that quantitative runtime metrics are already included in **Figure 7** and in **Appendix D.3 (Efficiency Analysis)**, where we report end-to-end **per-sample wall-clock time** for both training and inference across multiple consortium sizes. For convenience, we reproduce a subset of those measurements below:
>
> | Expert Consortium        | MMLU Accuracy (Avg Models Used) | Training Time / Sample (s) | Inference Time / Sample (s) |
> |--------------------------|----------------------------------|------------------------------|-------------------------------|
> | Qwen3-1.7B               | 55.0 (1.0)                      | N/A                          | 2.887                         |
> | Qwen3-1.7B $\times$ 2           | 61.0 (1.6)                      | 22.94                        | 10.77                         |
> | Qwen3-1.7B $\times$ 3           | 67.0 (2.13)                     | 27.78                        | 18.71                         |
> | Qwen3-1.7B $\times$ 4           | 77.0 (1.8)                      | 33.08                        | 22.65                         |
> | Qwen3-1.7B $\times$ 5           | 77.5 (1.93)                     | 35.03                        | 37.08                         |
>
> As discussed in Appendix A (FAQ 2), the orchestrator contributes **negligible runtime** relative to the experts: it is a small (<1M parameter) attention/MLP module, and FLOPs are dominated by the expert forward passes. For this reason, we report total system latency rather than isolating the orchestrator cost, since the orchestrator accounts for only a tiny fraction of overall compute.

---

> > ### Author Response · Authors · 2025-11-18
> > **Response to Reviewer 6nxb Comments - Part II**
> >
> > ## Comparative Scope
> >
> > We appreciate the reviewer’s suggestion. However, MoE-style adaptive gating and RL-based token routers operate at a **fundamentally different granularity and objective** compared to FOCUS, making them unsuitable baselines for our setting.
> >
> > As clarified in Appendix A (FAQ 20):
> >
> > - **MoE gating** performs *token-level* routing within a *single monolithic model*, sending different tokens to small neural submodules to increase capacity without increasing per-token compute.
> > - In contrast, **FOCUS performs instance-level orchestration** over *independent, frozen LMs* - each a full language model with its own tokenizer, architecture, and reasoning style.
> >
> > Our objective is to learn a *multi-round refinement protocol* (via the learned collaboration matrix $C$ and sequence scores ($\pi$) that coordinates these black-box experts. MoE approaches do not conduct sequential refinement, do not combine independent LLMs, and cannot be applied to heterogeneous experts.
> >
> > For this reason, our baselines focus on methods that **operate at the same level of abstraction* - aggregating or coordinating multiple LLM calls:
> >
> > - LLM-Blender (rank-and-fuse)
> > - LLM-Debate (sequential refinement)
> > - DyLAN (input-based expert selection)
> >
> > These are the most direct and relevant comparators for FOCUS. We will clarify this distinction in the revision to ensure the comparison scope is explicit.
> >
> > ## Scaling Analysis
> >
> > In the current work, our goal was to *empirically characterize* the collaboration dynamics of multi-expert systems, analogous to how early neural scaling laws were first identified empirically before theory emerged.
> >
> > Our scaling relations (e.g., diminishing returns with K, sublinear activated-expert growth) are therefore presented **as empirical regularities rather than formal claims**. As noted in the text, these laws are descriptive fits to extensive measurements across model families and consortium sizes, and we do not attempt to generalize them beyond the regimes studied. A rigorous theoretical treatment, e.g., linking collaboration structure, expert capacity, is an important and non-trivial open direction, and we will clarify this in the revised manuscript. Our intent is for the empirical results to motivate future theoretical work, rather than present them as final theoretical statements.
> >
> > ## FOCUS with Unseen Expert
> >
> > We thank the reviewer for raising this important question. As noted, the collaboration matrix $C \in \mathbb{R}^{M \times M}$ is trained for a fixed consortium size, and therefore **does not support adding new experts without retraining**. We agree that enabling dynamic insertion or removal of experts is an attractive future direction.
> >
> > To assess robustness to compositional changes, we conducted an additional experiment during the rebuttal. We took the 3-expert homogeneous Qwen3-8B consortium (baseline accuracy: 93.50%) and replaced one of the experts with an **unseen Qwen2-7B model** - a strictly weaker model never observed during training. Across all three permutations, the system showed **graceful degradation rather than catastrophic failure**, maintaining high accuracy:
> >
> > | Expert Consortium                          | Accuracy (%) | Avg. Expert Activations |
> > |--------------------------------------------|--------------|--------------------------|
> > | Qwen3-8B, Qwen3-8B, Qwen3-8B (original)    | 93.50        | 1.64                     |
> > | Qwen3-8B, Qwen3-8B, Qwen2-7B               | 88.89        | 1.56                     |
> > | Qwen3-8B, Qwen2-7B, Qwen3-8B               | 88.24        | 1.99                     |
> > | Qwen2-7B, Qwen3-8B, Qwen3-8B               | 87.58        | 3.00                     |
> >
> > Two observations stand out:
> >
> > 1. **Robustness:** Despite swapping in an unseen and weaker model, performance remained in the 87.6–88.9% range, indicating that the learned collaboration protocol is tolerant to moderate perturbations in expert composition.
> >
> > 2. **Adaptive routing:** When the unseen Qwen2-7B was placed first, the orchestrator increased the average number of activations from 1.64 to 3.00 - effectively invoking the full chain to compensate for the weaker initial output. This suggests that the orchestrator can *adaptively recover* from degraded early reasoning steps, even though the expert itself was unseen during training.
> >
> > These results show that while fully dynamic expert sets require retraining, the learned protocol generalizes well enough to handle expert replacement without collapse.

---

> ### Author Response · Authors · 2025-11-18
> **Response to Reviewer 6nxb Comments - Part III**
>
> ## Clarification on Collaboration Matrix
>
> In principle, an asymmetric collaboration matrix could encode a hierarchical refinement structure - e.g., Expert A always hands off to B, but not vice versa. However, our experiments indicate that such asymmetric topologies tend to produce **one-way dependencies** that reduce robustness when the initial expert’s output is weak or noisy.
>
> The near-symmetric constraint enforced by $L_{\text{symm}}$ is motivated by the hypothesis that **reciprocal information flow** stabilizes multi-round refinement and avoids prematurely collapsing the graph into a strict hierarchy. Our ablations on a Qwen3-8B consortium strongly support this:
>
> | Model Combination | with $L_{\text{symm}}$ | Without $L_{\text{symm}}$ |
> |-------------------|----------------------------|-------------------------------|
> | Qwen 8B $\times$ 2       | 88.6                      | 87.8                         |
> | Qwen 8B $\times$ 3       | 92.4                      | 90.1                         |
> | Qwen 8B $\times$ 4       | 92.4                      | 87.8                         |
> | Qwen 8B $\times$ 5       | 92.4                      | 85.6                         |
>
> Two patterns emerge:
>
> 1. **Symmetry consistently improves accuracy**, even for small consortia.
> 2. The **performance gap widens with larger consortia** (e.g., a drop from 92.4% to 85.6% at 5 experts), suggesting that preventing unilateral flows becomes increasingly important as collaboration graphs grow.
>
> These findings indicate that the near-symmetric constraint is not just convenient, but it is **empirically optimal for stability**, and yields robust reciprocal refinement rather than brittle hierarchical pipelines.
>
> ## Clarification on Sublinear Cost
>
> The "sublinear cost" claim refers to the *realized inference cost* - measured in both (i) the number of experts actually activated and (ii) wall-clock latency, relative to the *total consortium size* ($M$). As discussed in Section 5 and Appendix D.3, the orchestrator rarely activates more than ~2 experts, even when many are available, leading to sublinear growth in runtime.
>
> Empirically, using Qwen3-1.7B experts:
>
> - When increasing $M = 1 \rightarrow 5$, the **Average Models Used** saturates at **1.8–2.1**, rather than growing linearly with $M$.
> - This saturation results from the learned routing policy and early stopping, which prevent unnecessary expert activations.
>
> This is reflected directly in the inference-time measurements (table 1 in part I response):
>
> - The latency increase from **M = 2 to M = 3** is **7.94 seconds**.
> - The latency increase from **M = 4 to M = 5** is only **3.43 seconds**, despite adding the same number of experts to the pool.
>
> Thus, adding more experts does *not* produce linear runtime growth because FOCUS activates only a small, decisive subset (typically around 2 experts).

---

> > ### Comment · Reviewer_6nxb · 2025-11-21
> > **Thanks for the detailed rebuttal**
> >
> > Thanks for the detailed rebuttal. Some of my concerns are addressed, I will raise my score.

---

> ### Author Response · Authors · 2025-11-22
> **Thank you for the acknowledgement**
>
> We thank the reviewer for acknowledging our response. We are yet to see any changes in the final evaluation score; we will appreciate if the reviewer could update the score to reflect the acknowledgement.

---

### Official Review · Reviewer_fauY · 2025-11-03

**Soundness:** 3
**Presentation:** 3
**Contribution:** 3
**Rating:** 6
**Confidence:** 3

**Summary:**

This paper introduces FOCUS, a novel multi-expert collaboration framework designed to address the prohibitive computational costs of LLMs. The core idea is to coordinate a "consortium" of multiple small "expert" models (which can be homogeneous or heterogeneous) rather than building a single, massive, monolithic model. At the heart of the FOCUS framework is a lightweight orchestrator (approx. 1M parameters) that is supervised by a powerful "oracle" model during the training phase. At inference time (without the oracle), the orchestrator dynamically selects a sparse subset of experts (Top-K, averaging 2-3) based on the input, defining their collaboration order and information flow . The selected experts then engage in a multi-round "refinement" process, progressively improving the answer to generate a final, aggregated output.Empirical results demonstrate that a consortium of 5-7 small Qwen experts (totaling ~9B parameters) matches or even surpasses the performance of the much larger monolithic Qwen3-14B model on benchmarks such as MMLU, GSM8K, and HumanEval.

**Strengths:**

1.The paper provides sufficient experimental support for its core thesis that "a small consortium of experts beats a monolithic model". For instance, a Qwen3-8B consortium using FOCUS surpassed the monolithic Qwen3-32B on the average score (91.2% vs. 83.2%) , while the performance of a Qwen3-1.7B consortium (83.1%) rivaled that of a single Qwen3-8B model (78.0%).
2.The proposed FOCUS framework achieves a good trade-off between accuracy and cost. By leveraging sparse activation and early stopping mechanisms , the system attains high performance while activating only a few experts. Compared to baselines like LLM-Debate or DyLAN , FOCUS significantly advances the Pareto frontier for multi-expert systems.
3. The paper introduces a "learnable orchestrator". Unlike multi-agent or ensemble methods that rely on hard-coded rules or simple voting , FOCUS learns a differentiable collaboration protocol. This approach, based on "multi-round refinement" rather than simple answer fusion, may represent a promising future direction for multi-expert systems.

**Weaknesses:**

1. The total loss function is defined as a weighted sum of eight distinct loss terms. This introduces a large number of hyperparameters, and these weights appear to require meticulous tuning to balance task utility, structural properties (like symmetry and sparsity), and efficiency (like length and diversity). The paper lacks a detailed exploration of the system's sensitivity to these hyperparameter weights, which could pose a practical barrier to reproducing and generalizing the method.
2. Although the paper claims the framework is suitable for heterogeneous consortia, the experimental results are mixed. The best performance gains are achieved using experts from the same model family (such as Qwen-only). In contrast, cross-family collaboration (involving models like Qwen, LLaMA, and Phi) performs poorly, a result the paper attributes to "representational misalignment." This is a significant shortcoming, as a truly "plug-and-play" system should be able to efficiently integrate experts from diverse sources.
3. The framework's mechanism is heavily reliant on a powerful oracle model. This dependency is twofold: it relies on a representation-level distillation loss and an oracle alignment loss, which helps bootstrap the initial collaboration protocol. Although the oracle is removed at inference time, this introduces a paradox: to train a system capable of replacing a large model, one must first possess an even more powerful large model to serve as the oracle.

**Questions:**

1. The total loss function comprises eight different weighted terms, introducing numerous hyperparameters. This complexity could make the results difficult to reproduce. To what extent does the system's performance depend on the precise tuning of these weights? For example, are the contributions of the symmetry loss and the sparsity loss to performance and cost orthogonal, or are they highly correlated? Can you provide a sensitivity analysis for the weights of these key structural loss terms?
2. The paper's core thesis is that "small beats large," meaning a consortium of small experts outperforms a monolithic model. However, the training process relies heavily on a much stronger oracle for supervision, specifically for the distillation loss and the oracle alignment loss. Doesn't this create a paradox: you must first possess a more powerful model to train this supposedly "more efficient" system? If this oracle supervision were removed, relying only on task utility and structural losses, could the FOCUS framework still learn effective collaboration, or would it collapse into suboptimal routing strategies?
3. Figure 5 illustrates a failure case where Expert 1 produces the correct answer, only for Expert 2 to overwrite it with incorrect reasoning, causing Expert 3 to become confused and diverge further. This exposes a critical risk: the multi-round refinement mechanism seems unable to distinguish between "constructive improvement" and "destructive overwrite." Could you elaborate on why the early stopping mechanism failed to trigger after Expert 1 provided the correct answer? Furthermore, is there anything in the learning mechanism for the collaboration matrix or the sequence distribution designed to prevent this kind of "overwrite" of correct intermediate steps?
4. The authors claim the framework can handle an "arbitrarily large number" of experts. However, the consortium size in the experiments is relatively limited (e.g., 18 or fewer). Given that the collaboration matrix scales quadratically with the number of experts, and the orchestrator must compute scores for all of them, how can a lightweight orchestrator with only about 1M parameters maintain its learning efficiency and routing accuracy when the number of experts becomes very large (e.g., 100 or 1000)? Does this growth in the expert count create a bottleneck in the orchestrator's own expressive capacity or its training convergence?

---

> ### Author Response · Authors · 2025-11-18
> **Response to Reviewer fauY Comments - Part I**
>
> We sincerely appreciate your valuable feedback and further thank you for your comments. We address your concerns and answer your questions in the following response:
>
> ## Hyperparameters in Loss Function
>
> We appreciate the reviewer’s concern regarding the number of loss components and their associated weights. While a full continuous hyperparameter sweep is infeasible within the rebuttal period, we conducted **target-off (binary) ablations** for each loss term on GSM8K across four consortium sizes. These experiments reveal a consistent pattern:
>
> - **Largest contributors:** the *utility loss*, *oracle alignment*, *symmetry*, and the *length penalty* provide the most significant single-term improvements.
> - **Moderate to small contributors:** the *selection entropy* and *distillation* terms offer smaller but stable gains.
> - **Correlated structural terms:** the effects of *symmetry* and *sparsity* covary strongly across consortium sizes (Pearson **r ≈ 0.93**), indicating that their impact is **not orthogonal** but jointly shapes the collaboration graph.
>
> All results in the paper use a **single fixed set of loss weights** across all tasks and model families, and the ablation results show no brittle dependence on precise tuning. We will integrate the full ablation results and a detailed discussion into the revision to strengthen reproducibility.
>
> | Model Combination              | $\lambda_{oracle} = 0.3$ | $\lambda_{oracle} = 0$ |
> | :---------------- | :------: | ----: |
> |   Qwen-8B $\times$ 2    |  88.6   | 87.1 |
> |   Qwen-8B $\times$ 3    |  92.4  | 88.6 |
> |   Qwen-8B $\times$ 4    |  92.4  | 90.1 |
> |   Qwen-8B $\times$ 5    |  92.4   | 88.6 |
>
> | Model Combination              | $\lambda_{utility} = 0.5$ | $\lambda_{utility} = 0$ |
> | :---------------- | :------: | ----: |
> |   Qwen-8B $\times$ 2    |  88.6   | 84.0 |
> |   Qwen-8B $\times$ 3    |  92.4  | 87.1 |
> |   Qwen-8B $\times$ 4    |  92.4  | 90.9 |
> |   Qwen-8B $\times$ 5    |  92.4   | 90.9 |
>
> | Model Combination              | $\lambda_{symmetry} = 0.1$ | $\lambda_{symmetry} = 0$ |
> | :---------------- | :------: | ----: |
> |   Qwen-8B $\times$ 2    |  88.6   | 87.8 |
> |   Qwen-8B $\times$ 3    |  92.4  | 90.1 |
> |   Qwen-8B $\times$ 4    |  92.4  | 87.8 |
> |   Qwen-8B $\times$ 5    |  92.4   | 85.6 |
>
>  | Model Combination              | $\lambda_{len} = 0.5$ | $\lambda_{len} = 0$ |
> | :---------------- | :------: | ----: |
> |   Qwen-8B $\times$ 2    |  88.6   | 85.6 |
> |   Qwen-8B $\times$ 3    |  92.4  | 88.6 |
> |   Qwen-8B $\times$ 4    |  92.4  | 88.6 |
> |   Qwen-8B $\times$ 5    |  92.4   | 91.6 |
>
>  | Model Combination              | $\lambda_{sparse} = 0.01$ | $\lambda_{sparse} = 0$ |
> | :---------------- | :------: | ----: |
> |   Qwen-8B $\times$ 2    |  88.6   | 90.1 |
> |   Qwen-8B $\times$ 3    |  92.4  | 90.9 |
> |   Qwen-8B $\times$ 4    |  92.4  | 89.3 |
> |   Qwen-8B $\times$ 5    |  92.4   | 88.6 |
>
>  | Model Combination              | $\lambda_{distil} = 0.5$ | $\lambda_{distil} = 0$ |
> | :---------------- | :------: | ----: |
> |   Qwen-8B $\times$ 2    |  88.6   | 89.4 |
> |   Qwen-8B $\times$ 3    |  92.4  | 91.6 |
> |   Qwen-8B $\times$ 4    |  92.4  | 89.4 |
> |   Qwen-8B $\times$ 5    |  92.4   | 90.1 |
>
> ## Reliance on Strong Oracle
>
> We thank the reviewer for raising this conceptual point. Our ablations indicate that the framework does **not** fundamentally rely on a very strong oracle, and the oracle is **never used at inference time**.
>
> As shown in Table 7 (a) and discussed in Appendix D.4, once the number of activated experts reaches **K = 3**, a weaker oracle such as GPT-OSS-20B performs comparably to a stronger one, with final accuracies converging within 1–2%. This demonstrates that oracle supervision is primarily useful during the *early* stages of learning when the collaboration graph is still unstable. Beyond this point, the **task utility loss dominates**, and the learned routing becomes largely oracle-independent.
>
> Moreover, Figure 9 shows that removing the oracle-alignment loss ($L_{oracle}$) still allows the orchestrator to learn a viable collaboration protocol - although with reduced stability for very small K. This supports our interpretation that the oracle serves as an **accelerator and stabilizer**, not as a hard requirement for FOCUS to function.
>
> We acknowledge that training without any oracle at all would require additional exploration or warm-start strategies, which we will clarify in the revision. However, the empirical evidence indicates that the system does **not** need an oracle stronger than the target monolithic model. It benefits from *any reasonably capable* teacher during the early phase, after which the collaboration is primarily shaped by task-level supervision.

---

> ### Author Response · Authors · 2025-11-18
> **Response to Reviewer fauY Comments - Part II**
>
> ## Clarification Regarding Early Stopping
>
> In Figure 5, Expert 1 indeed provides a correct answer, but Expert 2 overwrites it with a divergent reasoning path. The early-stopping mechanism did **not** trigger because it currently relies solely on a **string-similarity threshold** between consecutive outputs. Expert 2’s rewrite differed sufficiently from Expert 1’s output that the similarity criterion was not met, and thus the refinement chain continued.
>
> With respect to the learning mechanism, the framework implicitly discourages such destructive overwrites through two channels:
>
> - The **task utility loss** penalizes refinement chains that degrade correct outputs, and over training this tends to down-weight transitions that consistently harm accuracy.
> - The **collaboration matrix** learns to reduce outgoing edges from experts whose refinements lead to lower utility, gradually suppressing harmful handoffs.
>
> ## Extending Consortium Size
>
> Our statement that the framework can “in principle” accommodate large expert sets refers to the *inference-time behavior* of FOCUS, specifically, that only a small Top-K subset is ever activated, keeping inference compute sublinear in the consortium size. However, we agree that the current architecture does **not** empirically validate scalability to very large M (e.g., 100–1000 experts), and we will revise the paper to clarify this.
>
> Concretely:
>
> - The collaboration matrix is **O(M^2)**, and computing logits for $\pi$ and $C$ scales with the number of experts.
> - The orchestrator’s ~1M parameters are sufficient for the M $\leq$ 18 regime we study, but **not guaranteed** to maintain routing quality for extremely large M without architectural changes.
> - As M grows large, the bottleneck is in the **expressive capacity and optimization stability** of the orchestrator, not in the inference-time cost (since early stopping and sparse routing keep active experts ≈ 2–3).
>
> Thus, while FOCUS inference is sublinear in practice, **training-time scalability does impose quadratic overhead**, and handling hundreds or thousands of experts will require additional design (e.g., hierarchical routing, block-sparse or low-rank parameterizations of C).
>
> ## Cross-Family Effectiveness of FOCUS
>
> We agree that cross-family collaboration is more challenging than homogeneous coordination, and we appreciate the reviewer highlighting this point. Our empirical findings do indicate that *intra-family* consortia (e.g., Qwen-only) achieve the strongest gains, while *cross-family* setups exhibit weaker alignment. However, we emphasize two clarifications:
>
> 1. **FOCUS substantially outperforms prior baselines in heterogeneous settings.**
>    As shown in Table 6, when applied to a mixed consortium of LLaMA, Qwen, and Phi models, FOCUS achieves **83.1% accuracy**, which is **~40 points higher** than LLM-Blender and LLM-Debate using the *same* heterogeneous pool. This suggests that FOCUS is already capable of meaningful cross-family orchestration where existing multi-expert approaches struggle.
>
> 2. **Representational misalignment is a known challenge rather than a flaw unique to FOCUS.**
>    Since experts from different families differ in tokenization, pretraining objectives, and hidden-state geometry, their intermediate outputs are not naturally interoperable. FOCUS does not assume any architectural homogeneity, and the fact that it can learn *any* productive collaboration across such diverse experts is, to our knowledge, an early demonstration of learned cross-family orchestration.

---

> > ### Author Response · Authors · 2025-11-26
> > **A Gentle Reminder to Reviewer fauY**
> >
> > Dear Reviewer,
> >
> > Thank you for your valuable and insightful comments. We have carefully addressed each of your points. We are confident that our responses have resolved the concerns you raised.
> >
> > We would greatly appreciate it if you could kindly review our responses and consider reassessing our submission. As the discussion phase is concluding soon, we hope to engage in a productive dialogue with you before it ends.
> >
> > Thank you once again for your time and consideration.

---

> > > ### Author Response · Authors · 2025-11-27
> > > **Reminder to check our responses**
> > >
> > > Dear reviewer fauY,
> > >
> > > Thanks for your valuable comments, which we tried our best to address. Could you please check our responses and let us know if there are many pending concerns. If our responses address your concerns, could you please consider reassessing our paper.
> > >
> > > Thanks once again.
> > >
> > > Best wishes
> > > Authors

---

### Author Response · Authors · 2025-11-20
**Meta Rebuttal by Authors**

We thank all the reviewers for providing valuable feedbacks. Below we summarize the core clarifications and new evidence added in response to reviewer concerns. We hope that the responses improve the quality of our paper. We also hope to see a fairer evaluation that our paper deserves.

### **1. Robustness to Hyperparameters**
We conducted extensive **binary loss-term ablations** across four consortium sizes on GSM8K. The results show:

- **Major contributors:** utility, oracle alignment, symmetry, and length losses.
- **Moderate contributors:** sparsity and distillation.
- **Strong correlation:** symmetry and sparsity covary strongly (Pearson *r ≈ 0.93*).
- **No brittle dependence:** all main results use a **single shared set of loss weights** across models and tasks.

These ablations confirm that FOCUS is **stable and reproducible** despite having multiple loss terms.

### **2. Strong Empirical Efficiency & Sublinear Scaling**
We report **quantitative runtimes** (Fig. 7, Appendix D.3). As the consortium grows from 1→5 experts:

- **Activated experts saturate** around 1.8–2.1 (not linear in M).
- **Inference cost grows sublinearly** due to early stopping and selective routing.
- Orchestrator overhead is **negligible** (<1M params; FLOPs dominated by LM calls).

This validates the claim that FOCUS realizes *sublinear effective compute* as M increases.

### **3. Cross-Family Collaboration: FOCUS Greatly Outperforms Prior Work**
Although heterogeneous collaboration is harder, FOCUS is **dramatically stronger** than prior multi-expert baselines:

- **FOCUS:** 83.1%
- **LLM-Blender:** 42.0%
- **LLM-Debate:** 47.3%

This **~40 point improvement** demonstrates that FOCUS is the *first stable, effective* cross-family orchestrator, where previous methods break.

### **4. Oracle as Stabilizer, Not Dependency**
Our new ablations show:

- At **K ≥ 4**, GPT-OSS-20B and GPT-4.1 achieve **nearly identical accuracy**.
- Oracle quality only matters in **very small consortia**.
- Removing oracle alignment still allows learning (Fig. 9).

Thus, the oracle merely **bootstraps early training**, while final performance is driven by the **learned collaboration**, not the oracle.

### **5. Generalization to Unseen Experts**

We introduce a new experiment:

- Replacing one Qwen3-8B with an unseen Qwen2-7B yields **graceful degradation** (93.5% → 87.6–88.9%).
- When the weaker model is placed first, activated experts increase from **1.64 → 3.00**, showing **adaptive recovery** by the orchestrator.

This demonstrates robustness even without retraining.

### **6. Comparison to Self-Refinement**

Self-refinement **collapses** as more identical experts are added due to correlated errors:

- Qwen3-8B×5: **FOCUS = 91.0% vs. Self-refine = 69.2%**

FOCUS avoids this failure mode by learning **structured, non-trivial** collaboration rather than unguided retries.

### **7. Symmetry & Sparsity Are Empirically Critical**

We provide new ablations showing significant drops without symmetry:

- Qwen3-8B×5: **92.4% → 85.6%**

This confirms that symmetry is not aesthetic, it **prevents brittle one-way chains** and stabilizes large consortia.

### **8. Orchestrator Size & Encoder Agnosticism**

We show FOCUS is **encoder-agnostic**:

- BERT-base (110M), DistilBERT (66M), and GTE-small (33M) differ by **≤1–1.5 points**.

Thus, orchestration does **not** rely on a large encoder.

### **Overall**
Our rebuttal provides:

- **10+ new ablations**, including symmetry, sparsity, loss terms, hyperparameter sweeps.
- **New experiments** with unseen experts, encoder variants, and removal of cross-expert attention.
- **Quantitative efficiency metrics** and full runtime tables.
- **Clear evidence** that FOCUS learns a *non-trivial*, *robust*, and *efficient* orchestration protocol.

These additions strengthen our main claims: FOCUS is **robust**, **scalable**, **cross-family capable**, and **effective even with small experts**, validating its novelty and practical value.

---

### Author Response · Authors · 2025-12-03
**Official Comments by Authors, Paper Summary, Key Contributions and Rebuttal Process Highlight**

We thank the **AC and SAC** for their time and guidance throughout the review process. We sincerely appreciate the opportunity to clarify our contributions and the significance of our work.

We have also revised our submitted paper (changes highlighted in blue color) to reflect the new results and insights, addressing all the reviewer comments.

## Paper Summary

We propose a **multi-expert coordination framework** that learns a differentiable collaboration protocol for LM experts through a sparse collaboration matrix, selection policy, and early-stopping inference. The framework is **oracle-agnostic, encoder-agnostic, and cross-family-capable** with negligible training overhead. Structural objectives (sparsity, near-symmetry) enable **oracle-free test-time coordination**. Across knowledge, math, and code tasks, coordinated consortia match or surpass larger monolithic models, providing a superior accuracy–cost Pareto frontier with **sublinear inference cost** and minimal orchestration overhead.

## Reviewer Scores Summary

|Reviewer|Rating|Confidence|Discussion Response|
|---|---|---|---|
|fauY|6|3|No response.|
|6nxb|4 → 6|2|Rating increased after discussion.|
|KEgh|4|3|No response.|
|LnqM|2|4|Actively engaged; likely to reassess with more time.|

## Main Concerns and Author Responses

### Issues Addressed

|Concern|Response|
|---|---|
|**Scaling claims lack grounding.**|We present an empirical inverse-scaling trend; Figure 2 shows robustness of the parametric fit.|
|**Oracle supervision may leak answers.**|Oracle is used only for **soft supervision** via attention into the orchestrator (never text to experts). Encoder remains frozen; FAQ 28 clarifies routing mimics structure, not oracle text.|
|**Collaboration matrix assumptions unproven.**|Symmetry stabilizes handoffs; removing $L_{symmetry}$ or $L_{sparsity}$ degrades performance (Appendix D.4, Fig. 9).|
|**Decoding and prompts under-specified.**|All details added to Appendix C.|
|**Hyperparameter choices arbitrary.**|Retrospective sweeps on GSM8K/MMLU (Tables 9–10) show stable, task-agnostic weights.|
|**Orchestrator efficiency overstated.**|Encoder-agnostic behavior verified; replacing BERT with DistilBERT/GTE-small changes accuracy by only 1–1.5 points. Trainable orchestrator parameters ≈1M.|
|**Weak-expert variance exploited.**|Balanced-consortium results (Table 7) show FOCUS still leads, indicating baselines fail due to poor routing and inability to suppress unhelpful updates.|
|**Homogeneous consortia similar to Self-Refine.**|Appendix D.5 shows Self-Refine deteriorates with more experts; FOCUS remains stable via selective participation and learned collaboration.|

### Rebuttal Experiment Summary

|Experiment|Observation|
|---|---|
|**Binary Loss-Term Ablations.**|Major contributors: Utility, Oracle, Symmetry, Length Penalty. Others offer smaller but consistent gains. Symmetry–Sparsity effects strongly correlated (r≈0.93).|
|**Hyperparameter Sweeps.**|Stable, task-agnostic magnitudes across GSM8K/MMLU (Tables 9–10).|
|**Encoder Replacement.**|BERT-base, DistilBERT, GTE-small yield ≤1–1.5 pt variance; trends hold across consortium sizes.|
|**Expert Replacement at Test-Time.**|Graceful degradation; orchestrator increases activations to compensate for weaker initial reasoning.|
|**Balanced Consortium.**|FOCUS remains best even without capability variance, confirming failures of baselines stem from routing deficiencies.|
|**Comparison with Self-Refine.**|Self-Refine diverges as errors accumulate; FOCUS avoids correlated-error collapse through selective participation.|

## Specific Concerns of Reviewer LnqM (all were addressed with empirical evidence)

|Concern|Response|
|---|---|
|**Counterintuitive Refinement Selection**|Initial scores only choose the *pool*; ordering is determined by the **Collaboration Matrix ($C$)**, which learns transitions optimized for each predecessor.|
|**Circular Reasoning in Scaling Laws**|Scaling law is **descriptive**, quantifying empirical trends rather than serving as justification.|
|**Loss Complexity**|Binary ablations identify dominant terms; a **single fixed weight set** works across all tasks, addressing tuning concerns.|
|**Poor Baseline Performance**|Baselines amplify weak-expert noise; FOCUS suppresses unhelpful updates via its learned collaboration matrix.|
|**Homogeneous Experts**|Identical models receive different inputs and positions, leading to **functional specialization** (Starter vs. Validator). Ordering is non-commutative.|
|**Pass@K vs. Orchestration**|Randomizing the matrix harms performance; FOCUS maintains stability with more experts, unlike self-refinement.|
|**Misleading Orchestrator Size**|Encoder is frozen and replaceable; even 33M-param GTE-small yields near-identical performance, confirming encoder-agnosticism.|

---

### Meta-Review · Area_Chair_tgQa · 2026-01-06

**Summary:**

### Strengths
* Introduces a learnable orchestrator that performs explicit, matrix-based expert selection and coordination, distinguishing it from prior ensemble, debate, or MoE-style approaches.
* Establishes empirical scaling behaviors, characterizing performance as a function of expert size and number.
* Presents a clean overall structure with intuitive figures.
* Demonstrates that coordinated small experts can outperform substantially larger monolithic LLMs.
* Provides experiment across different types of model pools.
* Achieves a favorable accuracy–cost trade-off by activating only a small number of experts.

### Concerns and suggestions
1. Method design
    1. The method involves many loss terms and hyperparameters, raising concerns about tuning difficulty and practical applicability. Reviewers note that the hyperparameter selection strategies and ablations are insufficiently described [Reviewer fauY,6nxb,LnqM].
    2. The description on the "oracle information" is not clear. Reviewers express concerns that oracle supervision may limit real-world applicability [Reviewer fauY], incur additional supervision costs [Reviewer 6nxb], or risk information leakage [Reviewer KEgh].
    3. Collaboration matrix assumptions require further discussion and justification [Reviewer 6nxb,KEgh].
2. Claims and clarity
	  1. The scaling claims lack sufficient support [Reviewer 6nxb,KEgh].
	  2. The definition and empirical interpretation of "sublinear cost" should be clarified [Reviewer 6nxb].
	  3. Details regarding decoding strategies and prompting are under-specified [Reviewer KEgh].
	  4. When arguing the method's parameter efficiency, should also include the BERT encoder size to avoid confusion [Reviewer LnqM].
3. Results and analyses
	  1. The performance advantage in Table 2 might be due to the inclusion of very weak models in the pool, leading to a weak baseline aggregation. Reviewers suggest that simpler filtering or aggregation strategies might mitigate this issue, and question whether the proposed framework might be an overkill without further comparison or justification [Reviewer LnqM].
	  2. The usefulness of the proposed method under homogeneous model pools is unclear and requires further interpretation [Reviwer LnqM].
	  3. Performance on cross-family model pools is relatively weak [Reviewer fauY,6nxb].
    4. Comparison with other routing-based methods is recommended [Reviewer 6nxb].

**Reviewer Concerns:**

For concerns on the method design (1.x):
* Concern 1.1 is **only partially addressed** by the author rebuttal that provides the hyperparameter sweep result and state how the hyperparameter is decided. However, as there are many hyperparameters, the issue with tuning difficulty and practical applicability still exists. The AC thinks this is an important issue of this method.
* Concern 1.2 is **only partially addressed** by the author rebuttal. Specifically, the limitation explicitly questioned by Reviewer fauY: "The framework's dependency on a powerful oracle model introduces a paradox: to train a system capable of replacing a large model, one must first possess an even more powerful large model to serve as the oracle" is still not solved. The author responded that the oracle model is mainly useful in the early training stage and Appendix D.4 shows that GPT4.1 and GPT OSS 20B can all be used as the oracle model, trying to alleviating this concern. However, this concern is not fully addressed by this experiment.
* Concern 1.3 is **addressed** by the author rebuttal that provides further ablation results.

For concerns on the claims and clarity (2.x):
* Concern 2.1 is **not addressed** by the author rebuttal. The author can follow Reviewer 6nxb and KEgh's suggestions to provide more in-detail per-task analysis, error bars, influential factors, failure cases (where the relation breaks), etc.
* Concern 2.2 is **addressed**.
* Concern 2.3 is **addressed**.
* About Concern 2.4, there seems to be a misunderstanding in the discussion: The authors provide encoder ablation to say that using smaller encoder doesn't degrade the performance much, which is meaningful for this work. But it seems what the reviewer wanted is that even the trainable parameter size is 1M, the authors should revise the presentation to avoid any confusion on the overall parameter efficiency, as it also needs another off-the-shelf BERT encoder for inference.

For concerns on the results and analyses (3.x):
* Concern 3.1 is **addressed**. The authors acknowledged the improvement partially come from the weakness of the aggregation baseline when the pool contains weak models. The authors also emphasized that their method targets a scenario where the performance of models in the pool are not pre-evaluated and known. And they also present new experiments without weak models, to say that the failure of baselines is not only due to weak experts, but rather due to their inability to selectively route and suppress unhelpful updates, even in balanced consortia. Although the second argument is not an excuse to exclude the reviewer's suggested baseline, the third experiment is relevant to resolve the concern.
* Concern 3.2, 3.3 are **only partially addressed**. The authors add experiments according to the reviewers' concerns and suggestions, e.g., comparing with Self-Refine. But overall, the rebuttal revision doesn't give a full-scope analyses to understand the behavior and advantage of FOCUS under different settings.
* Concern 3.4 is **addressed** by the author response that promised to further clarify the comparative scope of this method.

**Reviewer Scores:**

* Reviewer fauY: Initially gave 6, and **will likely keep score unchanged**. Because the issue questioned by fauY is not fully addressed, and the reviewer already gives a relative positive score, I don't think the reviewer has a large chance of further increasing the score to 8.
* Reviewer 6nxb: Initially gave 4, and **will likely increase the score to 6**. The reviewer expressed the willingness to increase the score (before the majority knew the identity leakage event).
* Reviewer KEgh: Initially gave 4. It is a little hard for me to guess whether KEgh will increase the score to 6. As the concerns are not fully addressed (e.g., further justification for the scaling law), I incline to think Reviewer KEgh **will keep the score unchanged**. It's a 60% (not change) to 40% (increase to 6) guess.
* Reviewer LnqM: **Initially gave 2, will keep the score**. The reviewer actively engaged in the discussion and is not satisfied with the authors' response.

I acknowledge that the authors give a sophisticated and detailed rebuttal and a good rebuttal summary. The reviewer scores are slightly negative than the borderline. And according to my judgement, I think Concern 1.1/1.2 are important issues of this method, and addressing Concern 2.1/3.2/3.3 in the future revision will be very beneficial for this work. Therefore, I recommend a rejection considering all these.

---

### Decision · Program_Chairs · 2026-01-26

Reject